# Provably Efficient Model-Free Constrained RL with Linear Function Approximation

**Arnob Ghosh**
Electrical and Computer Engineering
The Ohio State University
Columbus, OH, USA
`ghosh.244@osu.edu`

**Xingyu Zhou**
Electrical and Computer Engineering
Wayne State University
Detroit, MI, USA
`xingyu.zhou@wayne.edu`

**Ness Shroff**
Electrical and Computer Engineering
The Ohio State University
Columbus, OH, USA
`shroff.11@osu.edu`

## Abstract

We study the constrained reinforcement learning problem, in which an agent aims to maximize the expected cumulative reward subject to a constraint on the expected total value of a utility function. In contrast to existing model-based approaches or model-free methods accompanied with a 'simulator', we aim to develop the *first model-free*, *simulator-free* algorithm that achieves a sublinear regret and a sublinear constraint violation even in *large-scale* systems. To this end, we consider the episodic constrained Markov decision processes with linear function approximation, where the transition dynamics and the reward function can be represented as a linear function of some known feature mapping. We show that $\tilde{\mathcal{O}}(\sqrt{d^3 H^3 T})$ regret and $\tilde{\mathcal{O}}(\sqrt{d^3 H^3 T})$ constraint violation bounds can be achieved, where $d$ is the dimension of the feature mapping, $H$ is the length of the episode, and $T$ is the total number of steps. Our bounds are attained without explicitly estimating the unknown transition model or requiring a simulator, and they depend on the state space only through the dimension of the feature mapping. Hence our bounds hold even when the number of states goes to infinity. Our main results are achieved via novel adaptations of the standard LSVI-UCB algorithms. In particular, we first introduce primal-dual optimization into the LSVI-UCB algorithm to balance between regret and constraint violation. More importantly, we replace the standard greedy selection with respect to the state-action function in LSVI-UCB with a soft-max policy. This turns out to be key in establishing uniform concentration for the constrained case via its approximation-smoothness trade-off. Finally, we also show that one can achieve an even zero constraint violation for large enough $T$ by trading the regret a little bit but still maintaining the same order with respect to $T$.

## 1 Introduction

In many practical applications of online reinforcement learning (RL) (e.g., financial regulations, safety), there exist additional constraints on the learned policy in the sense that it also needs to ensure that the expected total utility (cost, resp.) exceeds a given threshold (is below a threshold, resp.). Constrained RL problem is formulated as a constrained Markov Decision Process (CMDP), in which the celebrated exploration-exploitation trade-off in online RL becomes more challenging due to the additional need to find a balance between reward regret and constraint violations.

36th Conference on Neural Information Processing Systems (NeurIPS 2022).

To develop online sample-efficient algorithms for CMDPs, prior works have largely resorted to model-based approaches, where the control policy is constructed based on a learned model [1–7]. However, due to the explicit estimation and storage of the unknown transition model, model-based approaches often lead to large time and space complexities. These issues are exacerbated in the large state space. Hence, there are several recent works starting to investigate model-free algorithms for CMDPs, which directly update the value function or the policy without first estimating the model [8–10]. However, all of these works consider an easier setting compared to standard RL in that they assume access to a simulator [11] (a.k.a. a generative model [12]), which is a strong oracle that allows the agent to query arbitrary state-action pairs and return the reward and the next state, hence greatly alleviating the intrinsic difficulty of exploration in RL. To the best of our knowledge, [13] were the first to study model-free and simulator-free algorithms for CMDPs. The above mentioned work on model-free algorithm considers the finite-state tabular setting and the regret scales polynomially with the number of states. Thus, the result *would not be useful* for large-scale RL applications where the number of states could even be infinite. To address this curse of dimensionality, modern RL has adopted *function approximation* techniques to approximate the (action-)value function of a policy, which greatly expands the potential reach of RL, especially via deep neural networks. However, little is known for the performance guarantee of *model-free algorithms* in CMDPs beyond tabular settings, even in the case of linear function approximation. Motivated by this, we are interested in the following question:

*Can we achieve provably sample-efficient and model-free exploration for CMDPs beyond tabular settings (without a simulator)?*

**Contribution.** To answer the above question, we consider the episodic CMDPs with linear function approximation, where the transition dynamics and the reward function can be represented as a linear function of some known feature mapping. Our main contributions are as follows.

- We show that with a proper parameter choice, our proposed algorithm achieves $\tilde{\mathcal{O}}(\sqrt{d^3 H^3 T})$ regret and $\tilde{\mathcal{O}}(\sqrt{d^3 H^3 T})$ constraint violation bounds with a high probability, where $d$ is the dimension of the feature mapping, $H$ is the length of the episode, and $T$ is the total number of steps. We also show that it is in fact possible to achieve *zero* constraint violation by trading the regret a little bit while maintaining the same order with respect to $T$.

- Our bounds are attained without explicitly estimating the unknown transition model or *requiring a simulator*, and they depend on the state space only through the dimension of the feature mapping. To the best of knowledge, these sub-linear bounds are the first results for model-free, simulator-free online RL algorithms for CMDPs with function approximations. We even improve the bound ($\tilde{\mathcal{O}}(T^{0.8})$) proposed in the model-free finite state tabular setting by [13] (Table 1).

- We combine the primal-dual algorithm with the classic model-free, simulator-free LSVI-UCB algorithm [14] to balance between regret and constraint violations. This naturally leads to the construction of a new composite state-action function (i.e., $Q$-function), which is the sum of the $Q$-function for the reward and the $Q$-function for the utility weighted by the dual variable. Due to this new type of $Q$-function in CMDPs, a key challenge arises when establishing the value-aware uniform concentration, which lies at the heart of the performance analysis of model-free exploration. *More specifically, the standard greedy selection with respect to this composite $Q$-function fails in finding non-trivial covering number for the function class of individual value functions (i.e., $V$-function) for the reward and the utility respectively.* To address this fundamental issue, we instead adopt a soft-max policy by utilizing its nice property of approximation-smoothness trade-off via its parameter, i.e., temperature coefficient.

## 1.1 Related Work

Model-based RL algorithms have been proposed for the CMDP [1–3, 5–7]. Apart from [7], the rest considered tabular set-up. In the tabular model-based set-up, the best known regret and constraint violations achieved are $\tilde{\mathcal{O}}(\sqrt{|\mathcal{S}|^2|\mathcal{A}|T})$ where $|\mathcal{S}|$ and $|\mathcal{A}|$ are the dimensions of the state and action spaces respectively. Hence, such results can not cope up with the large state space. [7] considered linear kernel MDP whereas we consider linear MDP. These two are not the same in general. We describe the differences with the algorithm proposed in [7] in Section 4.1.

Model-free RL algorithms have also been proposed [8–10] to solve CMDP. However, all of the above require a generator model, which simulates from any state and action. [13] proposed a 'triple-Q'

Table 1: Regret and Constraint Violations on Episodic MDP for algorithms which do not use simulators. (LFA: Linear Function Approximation, MOD-FREE: Model-Free)

| ALGORITHM | LFA? | MOD-FREE? | REGRET | VIOLATIONS |
|-----------|------|-----------|--------|------------|
| OPDOP [7] | YES | × | $\tilde{\mathcal{O}}(\sqrt{d^2 H^5 T})$ | $\tilde{\mathcal{O}}(\sqrt{d^2 H^5 T})$ |
| OPT-PRIMALDUAL CMDP [+][1] | NO | × | $\tilde{\mathcal{O}}(\sqrt{H^5 |\mathcal{S}|^2 |\mathcal{A}| T})$ | $\tilde{\mathcal{O}}(\sqrt{H^5 |\mathcal{S}|^2 |\mathcal{A}| T})$ |
| OPTDUAL-CMDP [+][1] | NO | × | $\tilde{\mathcal{O}}(\sqrt{H^3 |\mathcal{S}|^2 |\mathcal{A}| T})$ | $\tilde{\mathcal{O}}(\sqrt{H^3 |\mathcal{S}|^2 |\mathcal{A}| T})$ |
| OPTPRESS-PRIMALDUAL [6] | NO | × | $\tilde{\mathcal{O}}(\sqrt{H^5 |\mathcal{S}|^3 |\mathcal{A}| T})$ | $O(1)$ |
| TRIPLE-Q [13] | NO | $\sqrt{}$ | $\tilde{\mathcal{O}}(H^{3.2} T^{0.8} \sqrt{|\mathcal{S}||\mathcal{A}|})$ | $0$ |
| **OUR APPROACH** | YES | $\sqrt{}$ | $\tilde{\mathcal{O}}(\sqrt{d^3 H^3 T})$ | $\tilde{\mathcal{O}}(\sqrt{d^3 H^3 T})^*$ |

[*] WE CAN REDUCE THE VIOLATION TO $0$ FOR LARGE ENOUGH $T$ (FINITE) WHILE MAINTAINING THE SAME ORDER OF REGRET WITH RESPECT TO $T$ (APPENDIX H).

[+] WE REPLACE $\rho$ ($\xi$ IN OUR PAPER) BY $\mathcal{O}(H)$ SIMILAR TO OUR PAPER AND $\mathcal{N}$ BY $|\mathcal{S}|$.

algorithm which does not require a 'simulator'. However, it only considered tabular setting. The regret bound shown in [13] is $\tilde{\mathcal{O}}(T^{0.8})$ which is far from the optimal for the model-based case $\tilde{\mathcal{O}}(\sqrt{T})$. Please see Table 1 to see our contribution compared to the state-of-the-art approaches. [15] proposed a RL algorithm for the scenario where a constraint needs to be satisfied at each step of an episode. We consider a constraint where the cumulative utility over the length of the episodes must exceed a threshold. Hence, the set of constraints is fundamentally different. The authors in [15] also assumed that a safe-action is known for each state which we do not assume in our setting.

## 2 Problem Formulation

We consider an episodic constrained MDP, denoted by $(\mathcal{S}, \mathcal{A}, \mathbb{P}, H, r, g)$ where $\mathcal{S}$ is the state space, $\mathcal{A}$ is the action space, $H$ is the fixed length of each episode, $\mathbb{P} = \{\mathbb{P}_h\}_{h=1}^H$ is a collection of transition probability measures, $r = \{r_h\}_{h=1}^H$ is a collection of reward functions, and $g = \{g_h\}_{h=1}^H$ is a collection of utility functions. We assume that $\mathcal{S}$ is a measurable space with possibly infinite number of elements, $\mathcal{A}$ is a finite action set. $\mathbb{P}_h(\cdot|x, a)$ is the transition probability kernel which denotes the probability to reach a state when action $a$ is taken at state $x$. $r_h : \mathcal{S} \times \mathcal{A} \to [0, 1]$, and $g_h : \mathcal{S} \times \mathcal{A} \to [0, 1]$ and are assumed to be deterministic. However, we can readily extend to settings when $r_h$ and $g_h$ are random.

Each episode $k \in [K]$ starts with the fixed state $x_1$. It can be readily generalized to the setting where $x_1$ is drawn from a distribution. Then at each step $h \in [H]$ in episode $k$, the agent observes state $x_h^k \in \mathcal{S}$, picks an action $a_h^k \in \mathcal{A}$, receives a reward $r_h(x_h^k, a_h^k)$, and a utility $g_h(x_h^k, a_h^k)$. The MDP evolves to $x_{h+1}^k$ that is drawn from $\mathbb{P}_h(\cdot|x_h^k, a_h^k)$. The episode terminates at step $H + 1$. Without loss of generality, we assume that $r_{H+1} = g_{H+1} = 0$. In this paper, we consider the *challenging* scenario where the agent only observes the bandit information $r_h(x_h^k, a_h^k)$ and $g_h(x_h^k, a_h^k)$ at the visited state-action pair $(x_h^k, a_h^k)$. The policy-space of an agent is $\Delta(\mathcal{A}|\mathcal{S}, H)$; $\{\{\pi_h(\cdot|\cdot)\}_{h=1}^H : \pi_h(\cdot|x) \in \Delta(\mathcal{A}), \forall x \in \mathcal{S}, \text{and } h \in [H]\}$. Here $\Delta(\mathcal{A})$ is the probability simplex over the action space. For any $x_h^k \in \mathcal{S}$, $k \in [K]$, and $h \in [H]$, $\pi_{h,k}(a_h^k|x_h^k)$ denotes the probability that the action $a_h^k \in \mathcal{A}$ is taken at episode $k$ when the state is $x_h^k$.

Let $V_{r,h}^\pi(x)$ denote the expected value of the total reward function starting from step $h$ and state $x$ when the agent selects action using the policy $\pi = \{\pi_h\}_{h=1}^H$

$$V_{r,h}^\pi(x) = \mathbb{E}_\pi \left[ \sum_{i=h}^H r_i(x_i, a_i) | x_h = x \right], \tag{1}$$

where $\mathbb{E}$ is taken with respect to the policy $\pi$ and the transition probability kernel $\mathbb{P}$. Let $Q_{r,h}^\pi(x, a)$ denote the expected value of the total reward starting from step $h$ and the state-action pair $(x, a)$ and follows the policy $\pi$ as

$$Q_{r,h}^\pi(x, a) = \mathbb{E}_\pi \left[ \sum_{i=h}^H r_i(x_i, a_i) | x_h = x, a_h = a \right]. \tag{2}$$

Similarly, we define the value function for the utility $V_{g,h}^\pi(x)$, and the action-value function for the utility $Q_{g,h}^\pi(x,a)$. We denote $V_{j,h}^\pi(x)$, and $Q_{j,h}^\pi(x,a)$ for $j = r, g$.

**Definition 1.** *For brevity, we denote* $\mathbb{P}_h V_{j,h+1}^\pi(x,a) = \mathbb{E}_{x' \sim \mathbb{P}_h(\cdot|x,a)} V_{j,h+1}^\pi(x')$ *for* $j = r, g$.

Using this notation, the Bellman's equation associated with the policy $\pi$ becomes

$$Q_{j,h}^\pi(x,a) = (r_h + \mathbb{P}_h V_{j,h+1}^\pi)(x,a) \tag{3}$$

Note that $V_{j,h}^\pi(x) = \langle \pi_h(\cdot|x), Q_{j,h}^\pi(x,\cdot) \rangle_{\mathcal{A}}$, where $\langle \pi_h(\cdot|x), Q_{j,h}^\pi(x,\cdot) \rangle_{\mathcal{A}} = \sum_{a \in \mathcal{A}} \pi_h(a|x) Q_{j,h}^\pi(x,a)$.

The objective of the learning agent is to find an optimal solution of the following problem

$$\text{maximize }_{\pi \in \Delta(\mathcal{A}|\mathcal{S},\mathcal{H})} V_{r,1}^\pi(x_1), \qquad \text{subject to } V_{g,1}^\pi(x_1) \geq b. \tag{4}$$

Note that even though we have only once constraint, it can be readily generalized to the scenario with multiple constraints. In order to avoid trivial solutions, we consider $b \in (0, H]$. We denote the optimal policy as $\pi^*$ which solves the above optimization problem. Since $\pi^*$ is obtained by having complete information, it is also denoted as *the best policy in the hindsight*.

Without any constraint information a priori, an agent can not know the policies that satisfy the constraint. Instead, we allow the policy to violate the constraint and minimize the regret while minimizing the total constraint violations over the $K$ episodes. Such an approach is also considered in the existing literature [1, 7, 9]. We now define the performance metric which we seek to minimize.

**Performance Metric.** Let the policy employed by the agent at episode $k$ be $\pi_k = [\pi_{1,k}, \ldots, \pi_{h,k}, \ldots, \pi_{H,k}]^T$. The performance metric we are considering is the following

$$\text{Regret}(K) = \sum_{k=1}^K V_{r,1}^{\pi^*}(x_1) - V_{r,1}^{\pi_k}(x_1)$$

$$\text{Violation}(K) = \left[ \sum_{k=1}^K (b - V_{g,1}^{\pi_k}(x_1)) \right]_+, \tag{5}$$

where $[z]_+ = \max\{z, 0\}$. The regret is defined as the difference between the total reward value by following the optimal policy $\pi^*$, and the total reward value obtained by following agent's policy $\pi_k$ at episode $k$ over $K$ episodes. The constraint violation is defined as the difference between the threshold value $Kb$ and the total utility function attained by following the policies over all the episodes $K$.

**Linear Function Approximation.** To handle a possible large number of states, we consider the following linear MDPs.

**Assumption 1.** *The CMDP is a linear MDP with feature map* $\phi : \mathcal{S} \times \mathcal{A} \to \mathbb{R}^d$, *if for any* $h$, *there exists* $d$ *unknown signed measures* $\mu_h = \{\mu_h^1, \ldots, \mu_h^d\}$ *over* $\mathcal{S}$ *such that for any* $(x, a, x') \in \mathcal{S} \times \mathcal{A} \times \mathcal{S}$,

$$\mathbb{P}_h(x'|x,a) = \langle \phi(x,a), \mu_h(x') \rangle \tag{6}$$

*and there exists vectors* $\theta_{r,h}, \theta_{g,h} \in \mathbb{R}^d$ *such that for any* $(x,a) \in \mathcal{S} \times \mathcal{A}$,

$$r_h(x,a) = \langle \phi(x,a), \theta_{r,h} \rangle \quad g_h(x,a) = \langle \phi(x,a), \theta_{g,h} \rangle$$

Assumption 1 adapts the definition of linear MDP [14, 16] to the constrained case. By the above definition, the transition model, the reward, and the utility functions are linear in terms of feature map $\phi$. We remark that despite being linear, $\mathbb{P}_h(\cdot|x,a)$ can still have infinite degrees of freedom since $\mu_h(\cdot)$ is unknown. Note that tabular MDP is part of linear MDP [14].

Note that [7, 17] studied another related concept known as linear kernel MDP. In the linear kernel MDP, the transition probability is given by $\mathbb{P}_h(x'|x,a) = \langle \psi(x',x,a), \theta_h \rangle$. In general, linear MDP and linear kernel MDPs are two different classes of MDP [17].

Similar to Proposition 1 in [14], we can show that for a linear MDP and for any policy $\pi$ there exists $\{w_{j,h}^\pi\}_{h=1}^H$ such that $Q_{j,h}^\pi(x,a) = \langle w_{j,h}^\pi, \phi(x,a) \rangle$ for any $(x,a,h) \in \mathcal{S} \times \mathcal{A} \times [H]$. We, thus, focus on linear action-value function.

**Dual problem and Slater's Condition.** We first introduce few notations which we will use throughout this paper.

**Definition 2.** $V_h^{\pi,Y}(\cdot) = V_{h,r}^\pi(\cdot) + Y V_{h,g}^\pi(\cdot)$, and $Q_h^{\pi,Y}(x,a) = Q_{r,h}^\pi(x,a) + Y Q_{g,h}^\pi(x,a)$, where $Y$ is the dual variable.

Thus, $V_h^{\pi,Y}(\cdot)$ and $Q_h^{\pi,Y}(\cdot,\cdot)$ are respectively the composite value function and $Q$-functions respectively. We can cast the problem (4) as a saddle point problem $\max_\pi \min_Y \mathcal{L}(\pi, Y)$ where $\mathcal{L}(\pi, Y) = V_{r,1}^\pi(x_1) + Y(V_{g,1}^\pi(x_1) - b) = V_1^{\pi,Y} - Yb$, where $\pi$ is the primal policy and $Y$ is the dual variable. However, the lagrangian is non-concave in $\pi$ [18] even though it is convex in $Y$. Nevertheless, the strong duality holds [19]. Hence, there exists optimal dual variable $Y^*$, such that $\max_\pi \mathcal{L}(\pi, Y^*)$ will correspond to the optimal reward value function.

We assume the following slater's condition in this paper.

**Assumption 2** (Slater's Condition). *There exists $\gamma > 0$, and $\bar{\pi} \in \Delta(\mathcal{A}|\mathcal{D}, \mathcal{H})$, such that $V_{g,1}^{\bar{\pi}}(x_1) \geq b + \gamma$,*

**Lemma 1** (Boundedness of $Y^*$). *The optimal dual-variable $Y^* \leq \dfrac{V_{r,1}^{\pi^*}(x_1) - V_{r,1}^{\bar{\pi}}(x_1)}{\gamma} \leq \dfrac{H}{\gamma}$.*

The slater's condition is mild in practice and commonly adopted in previous works [7, 1, 20]. We use the properties of the slater's condition to bound the performance of our proposed algorithm.

**Definition 3.** *We set $\xi = 2H/\gamma$.*

## 3  Our Approach

---

**Algorithm 1** Model Free Primal-Dual Algorithm for Linear Function Approximation

---

1: **Initialization:** $Y_1 = 0$, $w_{j,h} = 0$, $\xi = 2H/\gamma$, $\alpha = \dfrac{\log(|\mathcal{A}|)K}{2(1+\xi+H)}$, $\eta = \xi/\sqrt{KH^2}$, $\beta = C_1 dH \sqrt{\log(4\log|\mathcal{A}|dT/p)}$

2: **for** episodes $k = 1, \dots, K$ **do**

3:   Receive the initial state $x_1^k$.

4:   **for** step $h = H, H-1, \dots, 1$ **do**

5:     $\Lambda_h^k \leftarrow \sum_{\tau=1}^{k-1} \phi(x_h^\tau, a_h^\tau)\phi(x_h^\tau, a_h^\tau)^T + \lambda \mathbf{I}$

6:     $w_{r,h}^k \leftarrow (\Lambda_h^k)^{-1}[\sum_{\tau=1}^{k-1} \phi(x_h^\tau, a_h^\tau)[r_h(x_h^\tau, a_h^\tau) + V_{r,h+1}^k(x_{h+1}^\tau)]]$

7:     $w_{g,h}^k \leftarrow (\Lambda_h^k)^{-1}[\sum_{\tau=1}^{k-1} \phi(x_h^\tau, a_h^\tau)[g_h(x_h^\tau, a_h^\tau) + V_{g,h+1}^k(x_{h+1}^\tau)]]$

8:     $Q_{r,h}^k(\cdot,\cdot) \leftarrow \min\{\langle w_{r,h}^k, \phi(\cdot,\cdot)\rangle + \beta(\phi(\cdot,\cdot)^T(\Lambda_h^k)^{-1}\phi(\cdot,\cdot))^{1/2}, H\}$

9:     $Q_{g,h}^k(\cdot,\cdot) \leftarrow \min\{\langle w_{g,h}^k, \phi(\cdot,\cdot)\rangle + \beta(\phi(\cdot,\cdot)^T(\Lambda_h^k)^{-1}\phi(\cdot,\cdot))^{1/2}, H\}$

10:     $\pi_{h,k}(a|\cdot) = \dfrac{\exp(\alpha(Q_{r,h}^k(\cdot,a) + Y_k Q_{g,h}^k(\cdot,a)))}{\sum_a \exp(\alpha(Q_{r,h}^k(\cdot,a) + Y_k Q_{g,h}^k(\cdot,a)))}$

11:     $V_{r,h}^k(\cdot) = \sum_a \pi_{h,k}(a|\cdot)Q_{r,h}^k(\cdot,a)$

12:     $V_{g,h}^k(\cdot) = \sum_a \pi_{h,k}(a|\cdot)Q_{g,h}^k(\cdot,a)$

13:   **for** step $h = 1, \dots, H$ **do**

14:     Compute $Q_{r,h}^k(x_h^k, a)$, $Q_{g,h}^k(x_h^k, a)$, $\pi(a|x_h^k)$ for all $a$.

15:     Take action $a_h^k \sim \pi_{h,k}(\cdot|x_h^k)$ and observe $x_{h+1}^k$.

16:   $Y_{k+1} = \max\{\min\{Y_k + \eta(b - V_{g,1}^k(x_1)), \xi\}, 0\}$

---

We now describe our proposed algorithm in Algorithm 1. This algorithm is based on the primal-dual adaptation of the LSVI-UCB [14]. For a given dual variable, the primal policy is updated, and then the dual value is updated based on the estimated utility value function. At each episode, the algorithm consists of three parts. The first part (Steps 4-12) consists of updating the parameters $w_{r,h}^k, w_{g,h}^k$ and $\Lambda_h^k$ which are used to update the $Q_{j,h}^k$ and $V_{j,h}^k$ at episode $k$. $\Lambda_h^k$ is the Gram-matrix for the regularized least square problem (see Eqn. (8), later). Note that the Steps 8-12 are not evaluated for each state, rather, they are evaluated only for the encountered states till episode $k-1$. Hence, we do not need to iterate over potentially infinite number of states. For the first episode, since $k-1=0$ and $\tau=1$, we have $w_{j,h}^k = 0, \forall j$ and $\Lambda_h^k = \lambda \mathbf{I}$. We note that $Q_{j,H+1}^k(\cdot,\cdot) = 0$ for $j = r, g$.

The value functions are updated (Steps 11-12) based on $Q$ function and the policy. The policy is based (Step 10) on a soft-max policy unlike the greedy one in the unconstrained case [14]. Soft-max policy $\text{SOFT-MAX}_\alpha(\mathbf{X}) = \{\text{SOFT-MAX}_\alpha^i(\mathbf{X})\}_{i=1}^{|\mathcal{A}|}$ for any vector $\mathbf{X} \in \mathbb{R}^{|\mathcal{A}|}$ is a $|\mathcal{A}|$-dimensional vector with parameter $\alpha$ where the $i$-th component

$$\text{SOFT-MAX}_\alpha^i(\mathbf{X}) = \frac{\exp(\alpha X_i)}{\sum_{n=1}^{|\mathcal{A}|} \exp(\alpha X_n)} \tag{7}$$

At step $h$, $\pi_{h,k}(a|x)$ is computed based on the soft-max policy on the composite $Q$-function vector $\{Q_{r,h}^k(x,a) + Y_k Q_{g,h}^k(x,a)\}_{a \in \mathcal{A}}$ where $Y_k$ is the lagrangian multiplier. When $\alpha = \infty$, this becomes equal to the greedy policy. The second part (Steps 13-15) is the execution of the soft-max policy based on the composite $Q$-value for the encountered state $x_h^k$.

**$Q$ function and Value function Estimation.** We need to estimate the value-function and $Q$-function with respect to the policy $\pi_k$. However, there are challenges. We do not know $\mathbb{P}_h$ in Bellman's equation (3), rather $\mathbb{P}_h V_{j,h+1}^{\pi_k}$ should be replaced by the empirical samples. Further, in the large state space, we can not iterate over all $(x,a)$. Rather, we parameterize $Q_{j,h}^{\pi^*}(\cdot, \cdot)$ by a linear form $\langle w_{j,h}^k(\cdot, \cdot), \phi(\cdot, \cdot) \rangle$. The intuition is to obtain $w_{j,h}^k$ from the Bellman's equation using the regularized least-square regression. We obtain $w_{j,h}^k$ for $j = r, g$ according to the following equation

$$w_{j,h}^k \leftarrow \arg\min_{w \in \mathbb{R}^d} \sum_{\tau=1}^{k-1} [j_h(x_h^\tau, a_h^\tau) + V_{j,h+1}^k(x_{h+1}^\tau) - w^T \phi(x_h^\tau, a_h^\tau)]^2 + \lambda ||w||_2^2 \tag{8}$$

Then, an additional bonus term $\beta(\phi(\cdot, \cdot)^T (\Lambda_h^k)^{-1} \phi(\cdot, \cdot))^{1/2}$ is added as in [14]. $\beta$ is constant which we will characterize in the next section. Such an additional term is used for upper confidence bound in LSVI-UCB [14]. The same additional term is used for both $Q_{r,h}^k$ and $Q_{g,h}^k$. Note the difference with the LSVI-UCB, here, we need to estimate the value function corresponding to the soft-max policy $\pi_k$ where in LSVI-UCB, a greedy policy corresponding to the $Q$-function is used.

**Policy.** We update a soft-max policy which selects actions according to the estimated 'composite' $Q$-function at the $k$-th episode. The reason behind using the soft-max policy instead of a greedy policy will be apparent in the next section when we state the main results and the proof ideas. Note from the strong duality, for optimal dual variable $Y^*$, optimal primal policy $\pi^*$ maximizes the composite value function $V^{\pi^*, Y^*}$ (Definition 2). Thus, the optimal policy should be a greedy one based on this optimal dual value $Y^*$. However, the greedy policy is not Lipschitz, hence, it does not provide uniform concentration bound for each individual value function, an essential step in the regret bound (Section 4.2). Hence, compared to the unconstrained scenario, we need more exploration in the policy space where the apparent reason is that we do not know the optimal dual variable beforehand. Since we use the soft-max policy, there is a gap compared to the optimal value even when the lagrangian multiplier $Y_k$ becomes equal to $Y^*$. However, if $\alpha$ in the soft-max policy also scales with $K$, then we can bound the gap from the optimal value function (Section 4.2).

**Dual Update.** To infer the constraint violation, we estimate $V_{g,1}^k$ for $V_{g,1}^{\pi_k}$. We update the lagrangian multiplier $Y_k$ by moving towards minimizing the lagrangian $\mathcal{L}(\pi, Y)$ over $Y \geq 0$ in line 16, where $\eta > 0$ is a step-size and $\xi$ is the upper bound on the dual variable such that optimal dual variable $Y^*$ is contained within $[0, \xi]$. The dual update is similar to the step described in [7, 1].

The dual update works as a trade-off between the reward maximization and the constraint violation reduction. If $b - V_{g,1}^k \geq 0$, that means with a high probability, the constraint will be violated for the policy $\pi_k$. Hence, the dual value is increased in order to focus on minimizing the constraint violation. Otherwise, the agent tries to maximize the reward value function.

**Space and Time Complexities.** We remark that Algorithm 1 only needs to store $Y_k$, $w_{r,h}^k, w_{g,h}^k, r_h(x_h^k, a_h^k), g_h(x_h^k, a_h^k), \Lambda_h^k$, and $\{\phi(x_h^k, a)\}_{a \in \mathcal{A}}$ for all $(h, k) \in [H] \times [K]$, hence, it takes $\mathcal{O}(d^2 H + d\mathcal{A}T)$ space. When we compute $(\Lambda_h^k)^{-1}$ using Sherman-Morrison formula, the computation of $V_{j,h+1}^k$ is dominated by computing $Q_{j,h+1}^k$ and the policy $\pi_k$. Hence, it takes $\mathcal{O}(d^2 \mathcal{A}T)$ time. Note that since our approach is model-free and we do not need to evaluate integrals as in [7] in order to estimate $\mathbb{P}_h V_{h+1}^k$.

Note that both $\eta$ and $\alpha$ use the knowledge of $K$. In case, $K$ is unknown, one can use the "doubling trick" [21] which will only scale the regret and constraint violation by a constant factor.

# 4 Analysis

We now state the main result. We prove that Algorithm 1 achieves regret and constraint violation which are sublinear in $T = KH$ where $T$ is the total number of steps.

## 4.1 Main Results

**Theorem 1.** *Fix $p > 0$. If we set $\lambda = 1$, $\beta = C_1 dH\sqrt{\iota}$ in Algorithm 1 where $\iota = \log(\log(|\mathcal{A}|)4dT/p)$ for some absolute constant $C_1$. With probability $(1 - p)$,*

$$\text{Regret}(K) \leq C\sqrt{d^3 H^3 T\iota^2} + \xi\sqrt{HT}$$

$$\text{Violation}(K) \leq \frac{C'2(1 + \xi)}{\xi}\sqrt{d^3 H^3 T\iota^2}$$

*for some absolute constants $C$, and $C'$.*

We remark the difference with the existing results. Since $\xi = 2H/\gamma$ (by Definition 3), our result indicates that our approach obtains $\tilde{\mathcal{O}}(\sqrt{d^3 H^3 T})$ regret and the same order of constraint violation where $\tilde{\mathcal{O}}$ absorbs logarithmic factor on $T$. The regret and constraint violation are sub-linear in $T$, and similar dependence is observed in [1, 7]. Also note that compared to the unconstrained case [14], there is an additional $\log(|\mathcal{A}|))$ factor in the value of $\iota$ which arises because we use soft-max policy instead of the greedy policy which adds to the covering number. The regret and constraint violation do not depend on the dimension of the state space, rather, it depends on the dimension of the feature space. *To the best of our knowledge this is the first result which shows both $\tilde{\mathcal{O}}(\sqrt{T})$ regret and constraint violation in the model-free set up (tabular or linear) without requiring a simulator.*

**Comparison with [7]**: Compared to [7] which also considers linear function approximation (however, it considers linear kernel MDP rather linear MDP) we improve the result in [7] by a factor of $H$. Second, compared to [7], which is a model-based policy-based algorithm, ours is a model-free value-based algorithm. Due to this, the above uniform concentration challenge does not exist in [7]. Moreover, our model-free algorithm also enjoys an easy implementation and improved computation efficiency since it does not estimate the next step expected value function as in [7] which requires an integration oracle to compute a $d$-dimensional integration at every step. [7] also needs to store the previous policies and estimated value functions, hence, it needs $\tilde{O}(T)$ additional space complexity. We have an additional $\sqrt{d}$ factor in front of the regret and constraint violation. Similar difference in regret is also observed between the model-based linear kernel unconstrained MDP [22] and model-free linear unconstrained MDP [14] *even in the unconstrained case*.

Similar to the discussion in Section 3.1 on [23], our result directly translates to a sample complexity guarantee (or, PAC guarantee). For example, we can learn a policy $\pi$ such that $V_{r,1}^{\pi^*}(x_1) - V_{r,1}^{\pi}(x_1) \leq \epsilon$, and $b - V_{g,1}^{\pi}(x_1) \leq \epsilon$ after $\tilde{\mathcal{O}}(d^3 H^4/\epsilon^2)$ number of samples. Here, the policy $\pi$ is obtained after running Algorithm 1 for $\tilde{\mathcal{O}}(d^3 H^3/\epsilon^2)$ number of episodes, and then selecting policy $\pi_k$ with probability $1/K$ for any $k \in [K]$.

Recently, [24] proposed an algorithm with provable sample complexity guarantee for linear CMDP. However, the regret and violation guarantees are different from the sample complexity guarantees as the former ones are *any time* guarantee. The proposed algorithms are different since the goal is different. In particular, the uniform concentration bound challenge does not appear there. Note that using the explore-then-commit algorithm [23], one can achieve $\tilde{\mathcal{O}}(T^{2/3})$ regret for large $T$ (from $\tilde{\mathcal{O}}(1/\epsilon^2)$ sample complexity bound achieved in [24]) which is worse than ours. Additionally, we achieve zero violation (Remark 2) while maintaining the same order of regret with respect to $T$.

## 4.2 Outline of the Proof

In this section, we provide an outline of our proof, which is mainly divided into three steps. We first establish a decomposition of the sum of reward regret and constraint violation. Then, we will bound two key terms that are related to optimism and prediction error, respectively. Finally, using standard optimization tools, we can achieve the main results. We highlight that the key challenges lie in the second step where a balance between the optimistic term and prediction error term is handled via the introduced soft-max policy.

**Step 1: Bounding the sum of Regret and violation scaled by dual variable** Similar to [1], we first establish the following decomposition, which upper bounds the sum of regret and violation. This will serve as the basis when applying optimization tools in Step 3.

**Lemma 2** (Decomposition). *For any $Y \in [0, \xi]$, we have*

$$\sum_{k=1}^{K}(V_{r,1}^{\pi^*}(x_1) - V_{r,1}^{\pi_k}(x_1)) + Y\sum_{k=1}^{K}(b - V_{g,1}^{\pi_k}(x_1)) \leq \frac{1}{2\eta}Y^2 + \frac{\eta}{2}H^2K +$$

$$\underbrace{\sum_{k=1}^{K}\left(V_{r,1}^{\pi^*}(x_1) + Y_k V_{g,1}^{\pi^*}(x_1)\right) - \left(V_{r,1}^{k}(x_1) + Y_k V_{g,1}^{k}(x_1)\right)}_{\mathcal{T}_1} +$$

$$\underbrace{\sum_{k=1}^{K}\left(V_{r,1}^{k}(x_1) - V_{r,1}^{\pi_k}(x_1)\right) + Y\sum_{k=1}^{K}\left(V_{g,1}^{k}(x_1) - V_{g,1}^{\pi_k}(x_1)\right)}_{\mathcal{T}_2}$$

Note that $\mathcal{T}_1$ is similar to the term related to optimism in the unconstrained case with the difference being that we now have two value functions weighted by the dual variable $Y_k$. Similarly, $\mathcal{T}_2$ is similar to prediction error term with the additional weight by $Y$. Since the first term in the above inequality can be easily bounded with a proper choice of $\eta$, we are only left to bound $\mathcal{T}_1$ and $\mathcal{T}_2$, respectively.

**Step 2: Bounding $\mathcal{T}_1$ and $\mathcal{T}_2$** To bound $\mathcal{T}_2$ and $\mathcal{T}_1$, we need to bound the difference between the *individual* estimated value function $V_{j,h}^{k}$ and the *individual* value function $V_{j,h}^{\pi}$ corresponding to a given policy $\pi$ at episode $k$. As in the unconstrained case, the key step is to control the fluctuations in least-squares value iteration. In particular, we need to show that for all $(k,h) \in [K] \times [H]$ with high probability

$$\left\| \sum_{\tau=1}^{k-1} \phi(x_h^\tau, a_h^\tau) \left[ V_{j,h+1}^{k}(x_{h+1}^\tau) - \mathbb{P}_h V_{j,h+1}^{k}(x_h^\tau, a_h^\tau) \right] \right\|_{(\Lambda_h^k)^{-1}}$$

is upper bounded by lower order term (e.g., $\mathcal{O}(d\sqrt{\log K})$). To this end, value-aware uniform concentration is required to handle the dependence between $V_{j,h+1}^{k}$ and samples $\{x_{h+1}^\tau\}_{\tau=1}^{k-1}$, which renders the standard self-normalized inequality infeasible in the model-free setting. The general idea here is to fix a function class $\mathcal{V}_{j,h}$ in advance and then show that each possible value function in our algorithm $V_{j,h}^{k}$ is within this class which has polynomial log-covering number. *In the following, we fix an $h \in [H]$ and drop the subscript $h$ for notation simplicity.*

**Uniform Concentration Bound for class of value function:** We first note that this uniform concentration bound is the main motivation for us to choose a soft-max policy as we will see that the standard greedy policy would fail in this case. That is, in order to guarantee that for each possible $V_j^k$, there is an $\epsilon$-close function in $\mathcal{V}_j$, it would basically lead to a very large covering number. To address this, we introduce soft-max policy and define the following corresponding function classes. We first define the following class for $Q$-function for $j = r, g$. $\mathcal{Q}_j = \{Q_j | Q_j(\cdot, \cdot) = \min\{\langle w_j, \phi(\cdot, \cdot)\rangle + \beta\sqrt{\phi(\cdot, \cdot)^T(\Lambda_h)^{-1}\phi(\cdot, \cdot)}, H\}\}$. Then, we define the following value function class $\mathcal{V}_j$. $\mathcal{V}_j = \{V_j | V_j(\cdot) = \sum_a \pi(a|\cdot)Q_j(\cdot, a); Q_j \in \mathcal{Q}_j, \pi \in \Pi\}$, where $\Pi$ is given by the following class $\Pi = \{\pi | \pi(a|\cdot) = \text{SOFT-MAX}_\alpha^a((Q_r(\cdot, \cdot) + YQ_g(\cdot, \cdot)); \forall a \in \mathcal{A}, Q_r \in \mathcal{Q}_r, Q_g \in \mathcal{Q}_g, Y \in [0, \xi]\}$, where SOFT-MAX is defined in (7).

**Why soft-max?** At this moment, we can explain why the introduction of soft-max in our algorithm is critical. Suppose we follow the standard greedy selection, which corresponds to $\alpha = \infty$ in above. The key issue in this approach is that one needs a large $\epsilon$-covering for $\mathcal{V}_j$ so that each possible $V_j^k$ can be well-approximated (i.e., $\epsilon$-close) by function in $\mathcal{V}_j$. This is in sharp contrast to the unconstrained case where $\mathcal{V}_j$ has a polynomial log-covering number. To see this difference, in the unconstrained case, we only have $V_r^k$ that is greedy with respect to $Q_r^k$. By the fact that $\max_a$ is a contraction map, an $\epsilon$-covering of the $Q_r^k$ implies an $\epsilon$-covering of $V_r^k$ and meanwhile the covering number of $\mathcal{Q}_r$ is reasonably small (Lemma 14) by standard arguments. This no longer holds in the constrained case due to the use of a composite $Q$-function. In particular, note that if the policy is greedy w.r.t. the composite $Q$-function, then an $\epsilon$-covering of $Q_j^k$ *fails* to be an

$\epsilon$-covering of $V_j^k$ in general since the greedy policy is not smooth in that a slight change of the composite $Q$-function could lead to a substantial change of the output action. This leads to a large distance for *individual* value functions due to the different action choices, even though the $Q$-function is close (Please see Appendix G for an example). Hence, one can not approximate individual value function within $\epsilon$-bound using greedy policy based on composite $Q$-function. This fact motivates us to turn to SOFT-MAX$_\alpha$, which is Lipschitz continuous with a Lipschitz constant at most $2\alpha$. Thus, our main idea is as follows. Given $Q_r^k$, $Q_g^k$ and $Y_k$, we can first find fixed $\tilde{Q}_r \in \mathcal{Q}_r$, $\tilde{Q}_g \in \mathcal{Q}_g$ and $\tilde{Y} \in [0, \xi]$ such that $\left\|Q_r^k - \tilde{Q}_r\right\|_\infty \leq \epsilon_1$, $\left\|Q_g^k - \tilde{Q}_g\right\|_\infty \leq \epsilon_2$, $\left\|Y^k - \tilde{Y}\right\|_\infty \leq \epsilon_3$ and $\left\|(Q_r^k + Y_k Q_g^k) - (\tilde{Q}_r + \tilde{Y}\tilde{Q}_g)\right\|_\infty \leq \epsilon$ with a reasonably small covering number. Then, thanks to the smoothness of soft-max function, we have $\|\pi_k - \tilde{\pi}\|_1 \leq 2\alpha\epsilon$ (Lemma 15). Combining this with the closeness of individual $Q$-function yields the closeness of individual $V$-function (Lemma 13). Hence, it ensures that the class $\mathcal{V}_j$ in our set-up has log-covering number of $\mathcal{O}(d\log(K))$.

**Choosing hyper-parameter $\alpha$ to achieve bound:** A larger value of $\alpha$ means that we need a smaller $\epsilon$, hence a larger covering number. Then, one may wonder if we can choose an arbitrarily small value for $\alpha$. However, the term $\mathcal{T}_1$ will be enlarged if we choose too small $\alpha$. Note that in the unconstrained case, $\mathcal{T}_1$ is upper bounded by zero due to optimism under greedy policy. Now, since we are using soft-max, we need to bound the approximation error between the soft-max and greedy one. As expected, in this case, a larger $\alpha$ leads to a smaller approximation error.

From the discussions above, we can see that the approximation-smoothness trade-off of the soft-max function is well captured by our $\mathcal{T}_1$ and $\mathcal{T}_2$, respectively. Therefore, we need to carefully choose the value of $\alpha$ to balance these two. In particular, with $\alpha = \frac{\log(|\mathcal{A}|)K}{2H}$, we have the following bounds on $\mathcal{T}_1$ and $\mathcal{T}_2$, respectively.

**Lemma 3.** *With probability $1 - p/2$, we have $\mathcal{T}_1 \leq K\frac{H\log(|\mathcal{A}|)}{\alpha}$. Hence, for $\alpha = \frac{\log(|\mathcal{A}|)K}{2(1+\xi+H)}$, we have $\mathcal{T}_1 \leq 2H(1+\xi+H)$ with probability $1 - p/2$.*

**Lemma 4.** *With probability at least $1 - p/2$, $\mathcal{T}_2 \leq \mathcal{O}((Y+1)\sqrt{d^3H^3T\iota^2})$, where $\iota = \log[\log(|\mathcal{A}|)4dT/p]$*

**Remark 1.** *The additional $\log|A|$ factor in $\iota$ arises as a trade-off for selecting soft-max policy as it is evident in Lemma 3. When we compensate by making $\alpha$ scaled with $\log|A|$ in Lemma 3, it increases the covering-number by $\log|A|$ as well in the $\iota$ term.*

**Step 3: Final Result by combining all the pieces:** By replacing $Y = 0$, $\eta = \xi/(\sqrt{KH^2})$, and combining Lemma 2,3, and 4, we obtain the regret bound. We also obtain the constraint violation using the idea from [1].

**Remark 2.** *We can reduce the violation to zero while maintaining the same order on regret with respect to $T$ (Appendix H). We consider a tighter optimization problem where we add $\zeta$ in the constraint of (4). In Appendix H, we bound the difference between the optimal value function for the tighter and the original problem as a function of $\zeta$. Since the tighter problem is also CMDP, we attain the regret and violation bound as in Theorem 1 with $b + \zeta$ in place of $b$. Hence, by choosing $\zeta$, we can show that it is possible to achieve $\tilde{\mathcal{O}}(\sqrt{T})$ regret and zero violation for large enough $K$ (Theorem 3 in Appendix H) albeit with an extra $H$ factor in front of regret bound.*

## 5 Experiments

We evaluate Algorithm 1 on a simulated model for job scheduling to validate our theoretical results. We consider that the number of jobs belongs to the discrete state $\{0, 1, \ldots, 9\}$ where $0$ means that there is no job. Total time horizon ($H$) is divided in 10 steps. After $H$ steps, a new episode begins. At the start of the each episode, the state of the job is assumed to be 9, i.e., the job stack is full. The agent needs to decide whether to send job ($a = 1$) or not ($a = 0$) to a machine. We assume that if $a = 1$, the agent sends 2 jobs to a machine and incurs a cost as the machine spends some resources to process the job. We assume that the rewards are step dependent. In particular, we assume that at time steps from 3 to 6, the reward is $1 - 0.9a$, In other time steps, the reward is $1 - 0.2a$. This mimics the setup where at certain time, it might be more costly to process a job (for example, electricity cost might be higher, or the machine needs to abandon an important job).

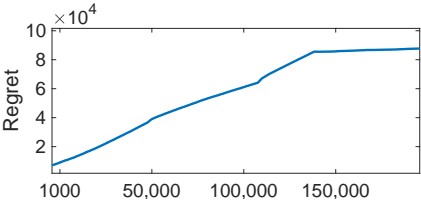 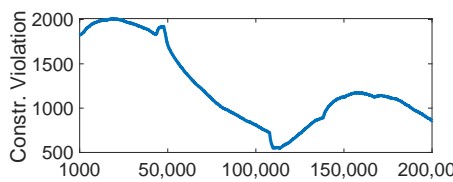

Figure 1: The plot for cumulative regret and constraint violation as a function of $K$. Each plot is an average of 10 trials.

We assume that even if the scheduler schedules a job, the machine might not be able to complete the 2 jobs. In particular,

$$x_{h+1} = \begin{cases} \max\{x_h - 2a, 0\} & \text{w.p. } 0.8 \\ \max\{x_h - a, 0\} & \text{w.p. } 0.1 \\ x_h, & \text{otherwise} \end{cases}$$

Thus, if $a = 0$, the state $x_{h+1} = x_h$. The agent gets an utility of $g(x_h, a_h, x_{h+1}) = (x_h - x_{h+1})/2$. We want that utility to be greater than or equal to $4$ at the end of every episode. This will ensure that at most 1 job can remain at the end of each episode.

We run Algorithm 1 for $2 \times 10^5$ episodes ($K$). The parameters we used are the followings: $\alpha = K/(1 + 2H/\gamma + H)$, $\eta = 2H/(\gamma \sqrt{KH^2})$. We set $\gamma = 1$. We have also set $\epsilon = 0.1$ in order to ensure that the violation goes towards 0 as $K$ increases. Note that the setup can be represented in a tabular form. Since linear MDP contains tabular form, the feature space representation becomes simple, in particular, $\phi(s, a) = \mathbf{e}_{s,a}$ where $\mathbf{e}_{s,a}$ is 1 for the state-action pair $(s, a)$, and 0 otherwise. The dimension of the feature space is $|\mathcal{S}||\mathcal{A}|$. The cumulative regret and constraint violations are shown in Figure 1. As predicted by our theory, the regret scales only as $\sqrt{K}$. On the other hand, the cumulative violation oscillates. However, the violation approaches 0 as $K$ increases. The oscillation of violation is due to the fact that dual variable also oscillates in order to illicit conservative response when the cumulative violation increases, and illicit aggressive response when the violation decreases.

## 6 Conclusion and Future Work

We propose a model-free RL-based algorithm for linear MDP. We have achieved $\tilde{\mathcal{O}}(\sqrt{d^3 H^3 T})$ regret and $\tilde{\mathcal{O}}(\sqrt{d^3 H^3 T})$ constraint violation. To the best of our knowledge, this is the first result which shows $\tilde{\mathcal{O}}(\sqrt{T})$ regret and $\tilde{\mathcal{O}}(\sqrt{T})$ constraint violation without requiring a generator for the model-free case. We have extended the LSVI-UCB algorithm in the primal-dual type framework. We have underlined the technical challenges in doing so and explained how the greedy policy fails to achieve an uniform concentration bound for individual value function. Subsequently, we show that a soft-max type algorithm achieves that.

Compared to [7], our regret is off by $\sqrt{d}$ factor which is due to the fact that we need to use uniform concentration bound. The similar gap is also observed for the unconstrained set-up as well. Whether we can tighten this dependence on $d$ remains an important future research direction. Whether we can tighten the dependence on $H$ also constitutes a future research direction. Extending the work to the setup where the feature-space needs to be learnt is also important. Recent works [25–27] on feature-space learning for unconstrained MDPs may provide some insights in this direction. Consideration of non-linear MDP also constitutes a future research direction.

**Acknowledgment**

XZ is partially supported by the NSF grant CNS-2153220. This work has been supported in part by NSF grants: 2112471 (also partly funded by DHS), CNS-2106933, and CNS-1901057 and a grant from the Army Research Office: W911NF-21-1-0244.

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
