# A  Preliminary Results

**Lemma 5.** *Under Assumption 1, for any fixed policy $\pi$, let $w_h^\pi$ be the corresponding weights such that $Q_{j,h}^\pi = \langle \phi(x,a), w_{j,h}^\pi \rangle$, for $j \in \{r,g\}$, then we have for all $h \in [H]$,*

$$||w_{j,h}^\pi|| \le 2H\sqrt{d} \tag{9}$$

*Proof.* From the linearity of the action-value function, we have

$$
\begin{aligned}
Q_{j,h}^\pi(x,a) &= j_h(x,a) + \mathbb{P}_h V_{j,h}^\pi(x,a) \\
&= \langle \phi(x,a), \theta_{j,h} \rangle + \int_{\mathcal{S}} V_{j,h+1}^\pi(x')\langle \phi(x,a), d\mu_h(x')\rangle \\
&= \langle \phi(x,a), w_{j,h}^\pi \rangle
\end{aligned}
\tag{10}
$$

where $w_{j,h}^\pi = \theta_{j,h} + \int_{\mathcal{S}} V_{j,h+1}^\pi(x')d\mu_h(x')$.

Now, $||\theta_{j,h}|| \le \sqrt{d}$, and $||\int_{\mathcal{S}} V_{j,h+1}^\pi(x')d\mu_h(x')|| \le H\sqrt{d}$. Thus, the result follows. $\qquad\square$

**Lemma 6.** *For any $(k,h)$, the weight $w_{j,h}^k$ satisfies*

$$||w_{j,h}^k|| \le 2H\sqrt{dk/\lambda} \tag{11}$$

*Proof.* For any vector $v \in \mathcal{R}^d$ we have

$$|v^T w_{j,h}^k| = |v^T(\Lambda_h^k)^{-1}\sum_{\tau=1}^{k-1}\phi_h^\tau(x_h^\tau, a_h^\tau)(j_h(x_h^\tau, a_h^\tau) + \sum_a \pi_{h+1,k}(a|x_{h+1}^\tau)Q_{j,h+1}^k(x_{h+1}^\tau, a))| \tag{12}$$

here $\pi_{h,k}(\cdot|x)$ is the Soft-max policy.

Note that $Q_{j,h+1}^k(x,a) \le H$ for any $(x,a)$. Hence, from (12) we have

$$
\begin{aligned}
|v^T w_{j,h}^k| &\le \sum_{\tau=1}^{k-1}|v^T(\Lambda_h^k)^{-1}\phi_h^\tau|.2H \\
&\le \sqrt{\sum_{\tau=1}^{k-1} v^T(\Lambda_k^h)^{-1}v}\sqrt{\sum_{\tau=1}^{k-1}\phi_h^\tau(\Lambda_h^k)^{-1}\phi_h^\tau}.2H \\
&\le 2H||v||\frac{\sqrt{dk}}{\sqrt{\lambda}}
\end{aligned}
\tag{13}
$$

Note that $||w_{j,h}^k|| = \max_{v:||v||=1}|v^T w_{j,h}^k|$. Hence, the result follows. $\qquad\square$

# B  Proof of Lemma 2

We first state and prove the following result which is similar to the one proved in [7].

**Lemma 7.** *For $Y \in [0, \xi]$,*

$$\sum_{k=1}^{K}(Y - Y_k)(b - V_{g,1}^k(x_1)) \leq \frac{Y^2}{2\eta} + \frac{\eta H^2 K}{2} \tag{14}$$

*Proof.*

$$\begin{aligned}
|Y_{k+1} - Y|^2 &= |Proj_{[0,\xi]}(Y_k + \eta(b - V_{g,1}^k(x_1))) - Proj_{[0,\xi]}(Y)|^2 \\
&\leq (Y_k + \eta(b - V_{g,1}^k(x_1))) - Y)^2 \\
&\leq (Y_k - Y)^2 + \eta^2 H^2 + 2\eta Y_k(b - V_{g,1}^k(x_1))
\end{aligned} \tag{15}$$

Summing over $k$, we obtain

$$0 \leq |Y_{K+1} - Y|^2 \leq |Y_1 - Y|^2 + 2\eta \sum_{k=1}^{K}(b - V_{g,1}^k(x_1))(Y_k - Y) + \eta^2 H^2 K$$

$$\sum_{k=1}^{K}(Y - Y_k)(b - V_{g,1}^k(x_1)) \leq \frac{|Y_1 - Y|^2}{2\eta} + \frac{\eta H^2 K}{2} \tag{16}$$

Since $Y_1 = 0$, we have the result. $\qquad \square$

Now, we prove Lemma 2.

*Proof.* Note that

$$Y \sum_{k=1}^{K}(b - V_{g,1}^{\pi_k}(x_1)) = \sum_{k}(Y - Y_k)(b - V_{g,1}^k(x_1)) + Y_k(b - V_{g,1}^k) + Y(V_{g,1}^k(x_1) - V_{g,1}^{\pi_k}(x_1))$$

$$\leq \frac{1}{2\eta}Y^2 + \frac{\eta}{2}H^2 K + \sum_{k=1}^{K}(Y_k b - Y_k V_{g,1}^k(x_1)) + Y(V_{g,1}^k(x_1) - V_{g,1}^{\pi_k}(x_1))$$

$$\leq \frac{1}{2\eta}Y^2 + \frac{\eta}{2}H^2 K + \sum_{k=1}^{K}(Y_k V_{g,1}^{\pi^*}(x_1) - Y_k V_{g,1}^k(x_1)) + \sum_{k=1}^{K}Y(V_{g,1}^k(x_1) - Y_k V_{g,1}^{\pi_k}(x_1))$$

where the first inequality follows from Lemma 7, and the second inequality follows from the fact that $V_{g,1}^{\pi^*}(x_1) \geq b$. Hence, the result simply follows from the above inequality. $\qquad \square$

# C  Proof of Lemmas 3 and 4

First, we prove some base results which we use to prove both Lemmas 3 and 4 in Section C.1. Subsequently, we prove Lemma 3 in Section C.2 and Lemma 4 in Section C.3.

## C.1  Proof of Base Results

We state and prove Lemmas 8,9, and 10.

First, we state the concentration lemma which is essential in controlling the fluctuations in the least square value iteration.

**Lemma 8.** *There exists a constant $C_2$ such that for any fixed $p \in (0, 1)$, if we let $\mathcal{E}$ be the event that*

$$\left\| \sum_{\tau=1}^{k-1} \phi_{j,h}^{\tau}[V_{j,h+1}^k(x_{h+1}^{\tau}) - \mathbb{P}_h V_{j,h+1}^k(x_h^{\tau}, a_h^{\tau})] \right\|_{(\Lambda_h^k)^{-1}} \leq C_2 d H \sqrt{\chi} \tag{17}$$

*for all $j \in \{r, g\}$, $\chi = \log[4(C_1 + 1)\log(|\mathcal{A}|)dT/p]$, for some constant $C_1$, then $\Pr(\mathcal{E}) = 1 - p/2$.*

The proof of Lemma 8 is technical and relegated to Appendix E. An extra $\log(|\mathcal{A}|)$ term appears because $\alpha$ appears in the covering number.

We now, recursively bound the difference between the value function maintained in Algorithm 1 (without the bonus term) and the value function for any policy for both the reward and utility value functions. We bound this using the expected difference at the next step plus an error term. This error term can be upper bounded by the bonus term with a high-probability.

**Lemma 9.** *There exists an absolute constant $\beta = C_1 dH\sqrt{\iota}$, $\iota = \log(\log(|\mathcal{A}|)4dT/p)$, and for any fixed policy $\pi$, on the event $\mathcal{E}$ defined in Lemma 8, we have*

$$\langle \phi(x,a), w_{j,h}^k \rangle - Q_{j,h}^\pi(x,a) = \mathbb{P}_h(V_{j,h+1}^k - V_{j,h+1}^\pi)(x,a) + \Delta_h^k(x,a) \tag{18}$$

*for some $\Delta_h^k(x,a)$ that satisfies $|\Delta_h^k(x,a)| \leq \beta\sqrt{\phi(x,a)^T(\Lambda_h^k)^{-1}\phi(x,a)}$.*

*Proof.* We only prove for $j = r$, the proof for $j = g$ is similar.

Note that $Q_{r,h}^\pi(x,a) = \langle \phi(x,a), w_{r,h}^\pi \rangle = r_h(x,a) + \mathbb{P}_h V_{r,h+1}^\pi(x,a)$.

Hence, we have

$$w_{r,h}^k - w_{r,h}^\pi = (\Lambda_h^k)^{-1}\sum_{\tau=1}^{k-1}\phi_h^\tau[r_h^\tau + V_{r,h+1}^k(x_{h+1}^\tau)] - w_{r,h}^\pi$$

$$= -\lambda(\Lambda_h^k)^{-1}(w_{r,h}^\pi) + (\Lambda_h^k)^{-1}\sum_{\tau=1}^{k-1}\phi_h^\tau[V_{r,h+1}^k(x_{h+1}^\tau) - \mathbb{P}_h V_{r,h+1}^k(x_h^\tau, a_h^\tau)]$$

$$+ (\Lambda_h^k)^{-1}\sum_{\tau=1}^{k-1}\phi_h^\tau[\mathbb{P}_h V_{r,h+1}^k(x_h^\tau, a_h^\tau) - \mathbb{P}_h V_{r,h+1}^\pi(x_h^\tau, a_h^\tau)] \tag{19}$$

Now, we bound each term in the right hand side of expression in (19). We call those terms as $\mathbf{q}_1$, $\mathbf{q}_2$, and $\mathbf{q}_3$ respectively.

First, note that

$$|\langle \phi(x,a), \mathbf{q}_1 \rangle| = |\lambda\langle \phi(x,a), (\Lambda_h^k)^{-1}(w_{r,h}^\pi) \rangle|$$

$$\leq \sqrt{\lambda}\|w_{r,h}^\pi\|\sqrt{\phi(x,a)^T(\Lambda_h^k)^{-1}\phi(x,a)} \tag{20}$$

Second, from Lemma 8, for the event in $\mathcal{E}$, we have

$$|\langle \phi(x,a), \mathbf{q}_2 \rangle| \leq CdH\sqrt{\chi}\sqrt{\phi(x,a)^T(\Lambda_h^k)^{-1}\phi(x,a)} \tag{21}$$

where $\chi = \log(4(C_1+1)\log(|\mathcal{A}|)dT/p)$. Third,

$$\langle \phi(x,a), \mathbf{q}_3 \rangle = \langle \phi(x,a), (\Lambda_h^k)^{-1}\sum_{\tau=1}^{k-1}\phi_h^\tau[\mathbb{P}_h(V_{r,h+1}^k - V_{r,h+1}^\pi)(x_h^\tau, a_h^\tau)] \rangle$$

$$= \langle \phi(x,a), (\Lambda_h^k)^{-1}\sum_{\tau=1}^{k-1}\phi_h^\tau(\phi_h^\tau)^T\int(V_{r,h+1}^k - V_{r,h+1}^\pi)(x')d\mu_h(x') \rangle$$

$$= \langle \phi(x,a), \int(V_{r,h+1}^k - V_{r,h+1}^\pi)(x')d\mu_h(x') \rangle - \langle \phi(x,a), \lambda(\Lambda_h^k)^{-1}\int(V_{r,h+1}^k - V_{r,h+1}^\pi)(x')d\mu_h(x') \rangle \tag{22}$$

The last term in (22) can be bounded as the following

$$|\langle \phi(x,a), \lambda(\Lambda_h^k)^{-1}\int(V_{r,h+1}^k - V_{r,h+1}^\pi)(x')d\mu_h(x') \rangle| \leq 2H\sqrt{d\lambda}\sqrt{\phi(x,a)^T(\Lambda_h^k)^{-1}\phi(x,a)} \tag{23}$$

since $\|\int(V_{r,h+1}^k - V_{r,h+1}^\pi)(x')d\mu_h(x')\|_2 \leq 2H\sqrt{d}$ as $\|\mu_h(\mathcal{S})\| \leq \sqrt{d}$. The first term in (22) is equal to

$$\mathbb{P}_h(V_{r,h+1}^k - V_{r,h+1}^\pi)(x,a) \tag{24}$$

Note that $\langle \phi(x,a), w_{r,h}^k \rangle - Q_{r,h}^\pi(x,a) = \langle \phi(x,a), w_{r,h}^k - w_{r,h}^\pi \rangle = \langle \phi(x,a), \mathbf{q_1} + \mathbf{q_2} + \mathbf{q_3} \rangle$. Since $\lambda = 1$, we have from (20), (21,(23), and (24)

$$|\langle \phi(x,a), w_{r,h}^k \rangle - Q_{r,h}^\pi(x,a) - \mathbb{P}_h(V_{r,h+1}^k - V_{r,h+1}^\pi)(x,a)| \le C_3 dH \sqrt{\chi} \sqrt{\phi(x,a)^T (\Lambda_h^k)^{-1} \phi(x,a)}$$
(25)

for some constant $C_3$ which is independent of $C_1$. Finally, note that

$$\begin{aligned}
C_3\sqrt{\chi} &= \sqrt{\log(4(C_1+1)\log(|\mathcal{A}|)dT/p)} \\
&= C_3\sqrt{\iota + \log(C_1+1)} \\
&\le C_1\sqrt{\iota}
\end{aligned}$$
(26)

where $\iota = \log(4\log(|\mathcal{A}|)dT/p)$. The last inequality follows from the fact that $\iota \in [\log 4, \infty)$ as $|A| \ge 2$, and $C_3$ is independent of $C_1$. Hence, we can always pick $C_3\sqrt{\log 4 + \log(C_1 + 1)} \le C_1\sqrt{\log 4}$ which satisfies (26) for all values of $\iota \in [\log 4, \infty)$. $\qquad\square$

Next, using the above lemma, we bound the difference between the composite value function maintained by the algorithm and the composite value function for a policy with the Lagrangian $Y_k$.

**Lemma 10.** *With prob.* $1 - p$, *(for the event in $\mathcal{E}$)*

$$Q_{r,h}^\pi(x,a) + Y_k Q_{g,h}^\pi(x,a) \le Q_{r,h}^k(x,a) + Y_k Q_{g,h}^k(x,a) - \mathbb{P}_h(V_{h+1}^k - V_{h+1}^{\pi,Y_k})(x,a)$$
(27)

*Proof.* From Lemma 9 and the fact that $|Q_{r,h}^\pi| \le H$, we have w.p. $1 - p/2$,

$$\begin{aligned}
Q_{r,h}^\pi(x,a) &\le \min\{\langle \phi(x,a), w_{r,h}^k \rangle + \beta\sqrt{\phi(x,a)^T(\Lambda_h^k)^{-1}\phi(x,a)}, H\} \\
&\quad + \mathbb{P}_h(V_{r,h+1}^\pi - V_{r,h+1}^k)(x,a) \\
&= Q_{r,h}^k(x,a) + \mathbb{P}_h(V_{r,h+1}^\pi - V_{r,h+1}^k)(x,a)
\end{aligned}$$

where the last equality follows from the definition of $Q_{r,h}^k$.

Similarly, with probability $1 - p/2$,

$$Y_k Q_{g,h}^\pi(x,a) \le Y_k Q_{g,h}^k(x,a) + Y_k \mathbb{P}_h(V_{g,h+1}^\pi - V_{g,h+1}^k)(x,a)$$

Hence, from union bound, with probability $1 - p$,

$$Q_{r,h}^\pi(x,a) + Y_k Q_{g,h}^\pi(x,a) \le Q_{r,h}^k + Y_k Q_{g,h}^k(x,a) + \mathbb{P}_h(V_{h+1}^{\pi,Y_k} - V_{h+1}^k)(x,a)$$

$\qquad\square$

## C.2 Proof of Lemma 3

First, we state and prove a supporting result which bounds the value functions corresponding to the greedy policy and the soft-max policy at a given step. We show that this gap can be controlled by the parameter $\alpha$.

**Lemma 11.** *Then,* $\bar{V}_h^k(x) - V_h^k(x) \le \dfrac{\log|\mathcal{A}|}{\alpha}$

where

**Definition 4.** $\bar{V}_h^k(\cdot) = \max_a[Q_{r,h}^k(\cdot,a) + Y_k Q_{g,h}^k(\cdot,a)]$.

$\bar{V}_h^k(\cdot)$ is the value function corresponds to the greedy-policy with respect to the composite $Q$-function.

*Proof.* Note that

$$V_h^k(x) = \sum_a \pi_{h,k}(a|x)[Q_{r,h}^k(x,a) + Y_k Q_{g,h}^k(x,a)]$$
(28)

where

$$\pi_{h,k}(a|x) = \frac{\exp(\alpha[Q_{r,h}^k(x,a) + Y_k Q_{g,h}^k(x,a)])}{\sum_a \exp(\alpha[Q_{r,h}^k(x,a) + Y_k Q_{g,h}^k(x,a)])} \tag{29}$$

Denote $a_x = \arg\max_a[Q_{r,h}^k(x,a) + Y_k Q_{g,h}^k(x,a)]$

Now, recall from Definition 4 that $\bar{V}_h^k(x) = [Q_{r,h}^k(x,a_x) + Y_k Q_{g,h}^k(x,a_x)]$. Then,

$$\bar{V}_h^k(x) - V_h^k(x) = [Q_{r,h}^k(x,a_x) + Y_k Q_{g,h}^k(x,a_x)]$$
$$- \sum_a \pi_{h,k}(a|x)[Q_{r,h}^k(x,a) + Y_k Q_{g,h}^k(x,a)]$$
$$\leq \left(\frac{\log(\sum_a \exp(\alpha(Q_{r,h}^k(x,a) + Y_k Q_{g,h}^k(x,a))))}{\alpha}\right)$$
$$- \sum_a \pi_{h,k}(a|x)[Q_{r,h}^k(x,a) + Y_k Q_{g,h}^k(x,a)]$$
$$\leq \frac{\log(|\mathcal{A}|)}{\alpha} \tag{30}$$

where the last inequality follows from Proposition 1 in [28]. $\square$

We are now ready to show Lemma 3.

*Proof.* We prove the lemma by Induction.

First, we prove for the step $H$.

Note that $Q_{j,H+1}^k = 0 = Q_{j,H+1}^\pi$.

Under the event in $\mathcal{E}$ as described in Lemma 8 and from Lemma 9, we have for $j = r, g$,

$$|\langle\phi(x,a), w_{j,H}^k(x,a)\rangle - Q_{j,H}^\pi(x,a)| \leq \beta\sqrt{\phi(x,a)^T(\Lambda_H^k)^{-1}\phi(x,a)}$$

Hence, for any $(x,a)$,

$$Q_{j,H}^\pi(x,a) \leq \min\{\langle\phi(x,a), w_{j,H}^k\rangle + \beta\sqrt{\phi(x,a)^T(\Lambda_H^k)^{-1}\phi(x,a)}, H\}$$
$$= Q_{j,H}^k(x,a) \tag{31}$$

Hence, from the definition of $\bar{V}_h^k$,

$$\bar{V}_H^k(x) = \max_a[Q_{r,H}^k(x,a) + Y_k Q_{g,h}^k(x,a)] \geq \sum_a \pi(a|x)[Q_{r,H}^\pi(x,a) + Y_k Q_{g,H}^\pi(x,a)]$$
$$= V_H^{\pi,Y_k}(x) \tag{32}$$

for any policy $\pi$. Thus, it also holds for $\pi^*$, the optimal policy. Hence, from Lemma 11, we have

$$V_H^{\pi^*,Y_k}(x) - V_H^k(x) \leq \frac{\log(|\mathcal{A}|)}{\alpha}$$

Now, suppose that it is true till the step $h+1$ and consider the step $h$.

Since, it is true till step $h+1$, thus, for any policy $\pi$,

$$\mathbb{P}_h(V_{h+1}^{\pi,Y_k} - V_{h+1}^k)(x,a) \leq \frac{(H-h)\log(|\mathcal{A}|)}{\alpha} \tag{33}$$

From (27) in Lemma 10 and the above result, we have for any $(x,a)$

$$Q_{r,h}^\pi(x,a) + Y_k Q_{g,h}^\pi(x,a) \leq Q_{r,h}^k(x,a) + Y_k Q_{g,h}^k(x,a) + \frac{(H-h)\log(|\mathcal{A}|)}{\alpha} \tag{34}$$

Hence,

$$V_h^{\pi, Y_k}(x) \le \bar{V}_h^k(x) + \frac{(H-h)\log(|\mathcal{A}|)}{\alpha} \tag{35}$$

Now, again from Lemma 11, we have $\bar{V}_h^k(x) - V_h^k(x) \le \frac{\log(|\mathcal{A}|)}{\alpha}$. Thus,

$$V_h^{\pi, Y_k}(x) - V_h^k(x) \le \frac{(H-h+1)\log(|\mathcal{A}|)}{\alpha} \tag{36}$$

Now, since it is true for any policy $\pi$, it will be true for $\pi^*$. From the definition of $V^{\pi, Y_k}$, we have

$$\left(V_{r,h}^{\pi^*}(x) + Y_k V_{g,h}^{\pi^*}(x)\right) - \left(V_{r,h}^k(x) + Y_k V_{g,h}^k(x)\right) \le \frac{(H-h+1)\log(|\mathcal{A}|)}{\alpha} \tag{37}$$

Hence, the result follows by summing over $K$ and considering $h = 1$. $\qquad\square$

## C.3  Proof of Lemma 4

In order to prove the Lemma 4, we state and prove the following result.

First, we introduce a notation. Let

$$D_{j,h,1}^k = \langle (Q_{j,h}^k(x_h^k, \cdot) - Q_{j,h}^{\pi_k}(x_h^k, \cdot)), \pi_{h,k}(\cdot | x_h^k) \rangle - (Q_{j,h}^k(x_h^k, a_h^k) - Q_{j,h}^{\pi_k}(x_h^k, a_h^k))$$
$$D_{j,h,2}^k = \mathbb{P}_h(V_{j,h+1}^k - V_{j,h+1}^{\pi_k})(x_h^k, a_h^k) - [V_{j,h+1}^k - V_{j,h+1}^{\pi_k}](x_{h+1}^k) \tag{38}$$

**Lemma 12.** *On the event defined in $\mathcal{E}$ in Lemma 8, we have*

$$V_{j,1}^k(x_1) - V_{j,1}^{\pi_k}(x_1) \le \sum_{h=1}^H (D_{j,h,1}^k + D_{j,h,2}^k) + \sum_{h=1}^H 2\beta\sqrt{\phi(x_h^k, a_h^k)^T (\Lambda_h^k)^{-1} \phi(x_h^k, a_h^k)} \tag{39}$$

*Proof.* By Lemma 9, for any $x, h, a, k$

$$\langle w_{j,h}^k(x, a), \phi(x, a) \rangle + \beta\sqrt{\phi(x, a)^T (\Lambda_h^k)^{-1} \phi(x, a)} - Q_{j,h}^{\pi_k}$$
$$\le \mathbb{P}_h(V_{j,h+1}^k - V_{j,h+1}^{\pi_k})(x, a) + 2\beta\sqrt{\phi(x, a)^T (\Lambda_h^k)^{-1} \phi(x, a)} \tag{40}$$

Thus,

$$Q_{j,h}^k(x, a) - Q_{j,h}^{\pi_k}(x, a) \le \mathbb{P}_h(V_{j,h+1}^k - V_{j,h+1}^{\pi_k})(x, a) + 2\beta\sqrt{\phi(x, a)^T (\Lambda_h^k)^{-1} \phi(x, a)}$$
$$\mathbb{P}_h(V_{j,h+1}^k - V_{j,h+1}^{\pi_k})(x, a) + 2\beta\sqrt{\phi(x, a)^T (\Lambda_h^k)^{-1} \phi(x, a)} - (Q_{j,h}^k(x, a) - Q_{j,h}^{\pi_k}(x, a)) \ge 0 \tag{41}$$

Since $V_{j,h}^k(x) = \sum_a \pi_{h,k}(a|x) Q_{j,h}^k(x, a)$ and $V_{j,h}^{\pi_k}(x) = \sum_a \pi_{h,k}(a|x) Q_{j,h}^{\pi_k}(x, a)$ where $\pi_{h,k}(a|\cdot) = \text{SOFT-MAX}_\alpha^a(Q_{r,h}^k + Y_k Q_{g,h}^k) \,\forall a$.

Thus, from (41),

$$V_{j,h}^k(x_h^k) - V_{j,h}^{\pi_k}(x_h^k) = \sum_a \pi_{h,k}(a|x_h^k)[Q_{j,h}^k(x_h^k, a) - Q_{j,h}^{\pi_k}(x_h^k, a)]$$

$$\le \sum_a \pi_{h,k}(a|x_h^k)[Q_{j,h}^k(x_h^k, a) - Q_{j,h}^{\pi_k}(x_h^k, a)]$$

$$+ 2\beta\sqrt{\phi(x_h^k, a_h^k)^T (\Lambda_h^k)^{-1} \phi(x_h^k, a_h^k)} + \mathbb{P}_h(V_{j,h+1}^k - V_{j,h+1}^{\pi_k})(x_h^k, a_h^k) - (Q_{j,h}^k(x_h^k, a_h^k) - Q_{j,h}^{\pi_k}(x_h^k, a_h^k)) \tag{42}$$

Thus, from (42), we have

$$V_{j,h}^k(x_h^k) - V_{j,h}^{\pi_k}(x_h^k) \le D_{j,h,1}^k + D_{j,h,2}^k + [V_{j,h+1}^k - V_{j,h+1}^{\pi_k}](x_{h+1}^k) + 2\beta\sqrt{\phi(x_h^k, a_h^k)^T (\Lambda_h^k)^{-1} \phi(x_h^k, a_h^k)} \tag{43}$$

Hence, by iterating recursively, we have

$$V_{j,1}^k(x_1) - V_{j,1}^{\pi_k}(x_1) \le \sum_{h=1}^H (D_{j,h,1}^k + D_{j,h,2}^k) + \sum_{h=1}^H 2\beta\sqrt{\phi(x_h^k, a_h^k)^T (\Lambda_h^k)^{-1} \phi(x_h^k, a_h^k)} \quad (44)$$

The result follows. □

We, are now ready to prove Lemma 4.

*Proof.* Note from Lemma 12, we have

$$\sum_{k=1}^K V_{j,1}^k(x_1) - V_{j,1}^{\pi_k}(x_1) \le \sum_{k=1}^K \sum_{h=1}^H (D_{j,h,1}^k + D_{j,h,2}^k) + \sum_{k=1}^K \sum_{h=1}^H 2\beta\sqrt{\phi(x_h^k, a_h^k)^T (\Lambda_h^k)^{-1} \phi(x_h^k, a_h^k)}$$
$$(45)$$

We, now, bound the individual terms. First, we show that the first term corresponds to a Martingale difference.

For any $(k, h) \in [K] \times [H]$, we define $\mathcal{F}_{h,1}^k$ as $\sigma$-algebra generated by the state-action sequences, reward, and constraint values, $\{(x_i^\tau, a_i^\tau)\}_{(\tau,i)\in[k-1]\times[H]} \cup \{(x_i^k, a_i^k)\}_{i\in[h]}$.

Similarly, we define the $\mathcal{F}_{h,2}^k$ as the $\sigma$-algebra generated by $\{(x_i^\tau, a_i^\tau)\}_{(\tau,i)\in[k-1]\times[H]} \cup \{(x_i^k, a_i^k)\}_{i\in[h]} \cup \{x_{h+1}^k\}$. $x_{H+1}^k$ is a null state for any $k \in [K]$.

A filtration is a sequence of $\sigma$-algebras $\{\mathcal{F}_{h,m}^k\}_{(k,h,m)\in[K]\times[H]\times[2]}$ in terms of time index

$$t(k, h, m) = 2(k-1)H + 2(h-1) + m \quad (46)$$

which holds that $\mathcal{F}_{h,m}^k \subset \mathcal{F}_{h',m'}^{k'}$ for any $t \le t'$.

Note from the definitions in (38) that $D_{j,h,1}^k \in \mathcal{F}_{h,1}^k$ and $D_{j,h,2}^k \in \mathcal{F}_{h,2}^k$. Thus, for any $(k, h) \in [K] \times [H]$,

$$\mathbb{E}[D_{j,h,1}^k | \mathcal{F}_{h-1,2}^k] = 0, \quad \mathbb{E}[D_{j,h,2}^k | \mathcal{F}_{h,1}^k] = 0 \quad (47)$$

Notice that $t(k, 0, 2) = t(k-1, H, 2) = 2(H-1)k$. Clearly, $\mathcal{F}_{0,2}^k = \mathcal{F}_{H,2}^{k-1}$ for any $k \ge 2$. Let $\mathcal{F}_{0,2}^1$ be empty. We define a Martingale sequence

$$M_{j,h,m}^k = \sum_{\tau=1}^{k-1} \sum_{i=1}^H (D_{j,i,1}^\tau + D_{j,i,2}^\tau) + \sum_{i=1}^{h-1} (D_{j,i,1}^k + D_{j,i,2}^k) + \sum_{l=1}^m D_{j,h,l}^k$$
$$= \sum_{(\tau,i,l)\in[K]\times[H]\times[2], t(\tau,i,l)\le t(k,h,m)} D_{j,i,l}^\tau \quad (48)$$

where $t(k, h, m) = 2(k-1)H + 2(h-1) + m$ is the time index. Clearly, this martingale is adopted to the filtration $\{\mathcal{F}_{h,m}^k\}_{(k,h,m)\in[K]\times[H]\times[2]}$, and particularly

$$\sum_{k=1}^K \sum_{h=1}^H (D_{j,h,1}^k + D_{j,h,2}^k) = M_{j,H,2}^K \quad (49)$$

Thus, $M_{j,H,2}^K$ is a Martingale difference satisfying $|M_{j,H,2}^K| \le 4H$ since $|D_{j,h,1}^k|, |D_{j,h,2}^k| \le 2H$ From the Azuma-Hoeffding inequality, we have

$$\Pr(M_{j,H,2}^K > s) \le 2\exp(-\frac{s^2}{16TH^2}) \quad (50)$$

With probability $1 - p/2$ at least for any $j = r, g$,

$$\sum_k \sum_h M_{j,H,2}^K \le \sqrt{16TH^2 \log(4/p)} \quad (51)$$

Now, we bound the second term. Note that the minimum eigen value of $\Lambda_h^k$ is at least $\lambda = 1$ for all $(k, h) \in [K] \times [H]$. By Lemma 17,

$$\sum_{k=1}^{K} (\phi_h^k)^T (\Lambda_h^k)^{-1} \phi_h^k \leq 2 \log \left[ \frac{\det(\Lambda_h^{k+1})}{\det(\Lambda_h^1)} \right] \tag{52}$$

Moreover, note that $||\Lambda_h^{k+1}|| = ||\sum_{\tau=1}^{k} \phi_h^k (\phi_h^k)^T + \lambda \mathbf{I}|| \leq \lambda + k$, hence,

$$\sum_{k=1}^{K} (\phi_h^k)^T (\Lambda_h^k)^{-1} \phi_h^k \leq 2d \log \left[ \frac{\lambda + k}{\lambda} \right] \leq 2d\iota \tag{53}$$

Now, by Cauchy-Schwartz inequality, we have

$$\sum_{k=1}^{K} \sum_{h=1}^{H} \sqrt{(\phi_h^k)^T (\Lambda_h^k)^{-1} \phi_h^k} \leq \sum_{h=1}^{H} \sqrt{K} [\sum_{k=1}^{K} (\phi_h^k)^T (\Lambda_h^k)^{-1} \phi_h^k]^{1/2}$$

$$\leq H \sqrt{2dK\iota} \tag{54}$$

Note that $\beta = C_1 d H \sqrt{\iota}$.

Thus, we have with probability $1 - p/2$,

$$\sum_{k=1}^{K} V_{r,1}^k(x_1) - V_{r,1}^{\pi_k}(x_1) + Y \sum_{k=1}^{K} (V_{g,1}^k(x_1) - V_{g,1}^{\pi_k}(x_1))$$

$$\leq (Y+1)[\sqrt{2TH^2 \log(4/p)} + C_4 \sqrt{d^3 H^3 T \iota^2}] \tag{55}$$

Hence, the result follows. $\qquad\square$

# D   Proof of Theorem 1

We, first, show the regret bound. Note from Lemma 2, Lemma 3, and Lemma 4, we have for $Y \in [0, \xi]$ at least w.p. $1 - p$,

$$\sum_{k=1}^{K} (V_{r,1}^{\pi^*}(x_1) - V_{r,1}^{\pi_k}(x_1)) + Y \sum_{k=1}^{K} (b - V_{g,1}^{\pi_k}(x_1))$$

$$\leq \frac{Y^2}{2\eta} + \frac{\eta}{2} H^2 K + \frac{HK \log|\mathcal{A}|}{\alpha} + \tilde{\mathcal{O}}((Y+1)\sqrt{d^3 H^3 T \iota^2}) \tag{56}$$

Replacing $Y$ with 0 in (56), we have

$$\sum_{k=1}^{K} (V_{r,1}^{\pi^*}(x_1) - V_{r,1}^{\pi_k}(x_1) \leq \frac{\eta}{2} H^2 K + \frac{HK \log|\mathcal{A}|}{\alpha} + \mathcal{O}(\sqrt{d^3 H^3 T \iota^2}) \tag{57}$$

By noting that $\eta = \frac{\xi}{\sqrt{KH^2}}$, and $\alpha = \frac{\log|\mathcal{A}|K}{2(1 + \xi + H)}$, we have the result.

We, now, show the violation bound. Note from Lemma 2, Lemma 3, and Lemma 4 that w.p. $1 - p$ (at least),

$$\sum_{k=1}^{K} (V_{r,1}^{\pi^*}(x_1) - V_{r,1}^{\pi_k}(x_1) + Y(b - V_{g,1}^{\pi_k}(x_1))) \leq \frac{1}{2\eta} Y^2 + \frac{\eta}{2} H^2 K + \frac{HK \log|\mathcal{A}|}{\alpha} +$$

$$(1 + Y)\mathcal{O}(\sqrt{H^3 T \iota^2}) \tag{58}$$

Now, put $\eta = \frac{\xi}{\sqrt{KH^2}}$, $\alpha = \frac{\log|\mathcal{A}|K}{2(1 + \xi + H)}$, and $Y \leq \xi$, then, we have

$$\sum_{k=1}^{K} (V_{r,1}^{\pi^*}(x_1) - V_{r,1}^{\pi_k}(x_1) + Y(b - V_{g,1}^{\pi_k}(x_1))) \leq (1 + \xi)\mathcal{O}(\sqrt{H^3 T \iota^2}) + \xi\sqrt{KH^2} \tag{59}$$

Now, there exists a policy $\pi'$ such that $V_{r,1}^{\pi'} = \frac{1}{K}\sum_{k=1}^K V_{r,1}^{\pi_k}$, $V_{g,1}^{\pi'} = \frac{1}{K}\sum_{k=1}^K V_{g,1}^{\pi_k}$. By the occupancy measure, $V_{r,1}^{\pi}$ and $V_{g,1}^{\pi}$ are linear in occupancy measure induced by $\pi$. Thus, the average of $K$ occupancy measure also produces an occupancy measure which induces policy $\pi'$ and $V_{r,1}^{\pi'}$, and $V_{g,1}^{\pi'}$. We take $Y = 0$ when $\sum_{k=1}^K (b - V_{g,1}^{\pi_k}(x_1^k)) < 0$, otherwise $Y = \xi$. Hence, we have

$$(V_{r,1}^{\pi^*}(x_1) - \frac{1}{K}\sum_{k=1}^K V_{r,1}^{\pi_k}(x_1) + \xi(b - \frac{1}{K}\sum_{k=1}^K V_{g,1}^{\pi_k}(x_1))_+$$

$$= (V_{r,1}^{\pi^*}(x_1) - V_{r,1}^{\pi'}(x_1) + \xi[b - V_{g,1}^{\pi'}(x_1)]_+$$

$$\leq \frac{(1+\xi)\mathcal{O}(\sqrt{d^3 H^3 T \iota^2})}{K} + \xi\frac{\sqrt{KH^2}}{K}$$

Since $\xi = 2H/\gamma$, and using the result of strong duality (Lemma 19), we have

$$(b - \frac{1}{K}\sum_{k=1}^K V_{g,1}^{\pi_k}(x_1^k))_+ \leq \frac{2(1+\xi)}{K\xi}\mathcal{O}(\sqrt{d^3 H^3 T \iota^2}) \tag{60}$$

Hence, the result follows.

# E   Proof of Lemma 8

To simplify the notation, we remove $h$ from the subscript from $Q_{j,h}^k$ and $V_{j,h}^k$ in this Section.

In order to prove the Lemma 8, we first compute the $\epsilon$-covering number for the class of value functions (Lemma 13). In order to compute that we first compute the $\epsilon$-covering number of the individual $Q$-functions (Lemma 14) which is essential to compute the covering number for composite $Q$-functions (Corollary 1). Subsequently, we show that if the two $Q$-functions and the Lagrange multipliers are close, the policies are also close (Lemma 15).

We first introduce the set of $Q$-functions.

**Definition 5.** *Let* $\mathcal{Q}_j = \{Q|Q(\cdot,\cdot) = \min\{w_j^T \phi(\cdot,\cdot) + \beta\sqrt{\phi^T(\cdot,\cdot)^T \Lambda^{-1}\phi(\cdot,\cdot)}, H\}\}$

The set $\mathcal{Q}$ is parameterized by $w_j$, and $\Lambda$. We have $||w_j|| \leq 2H\sqrt{dk/\lambda}$ (from Lemma 6). The minimum eigen value of $\Lambda$ satisfies $\lambda_{min} \geq 1$. Hence, the Frobenius norm of $\Lambda^{-1}$ is bounded. Note that $Q_j^k \in \mathcal{Q}_j$ for $j = r,g$.

We now introduce the class of value function for $j = r,g$.

**Definition 6.** *Let* $\mathcal{V}_j = \{V_j|V_j(\cdot) = \sum_a \pi(a|\cdot)Q_j(\cdot,a); Q_r \in \mathcal{Q}_r, Q_g \in \mathcal{Q}_j, Y \in [0,\xi]\}$ *for* $j = r,g$, *where*

$$\Pi = \{\pi|\forall a \in \mathcal{A}, \pi(a|\cdot) = \text{SOFT-MAX}_\alpha^a((Q_r(\cdot,\cdot) + YQ_g(\cdot,\cdot))Q_r \in \mathcal{Q}_r, Q_g \in \mathcal{Q}_g, Y \in [0,\xi]\}.$$

The class of value function $\mathcal{V}_j$ is parameterized by $w_r, w_g, \Lambda$, and $Y \in [0,\xi]$. Note that even the individual value function depends on the $Q$-functions for both the reward and utility since the policy depends on the composite $Q$-function.

First, we need to see whether $V_j^k \in \mathcal{V}_j$. Recall the definition of $V_j^k$ at the $k$-th episode $V_j^k(\cdot) = \sum_a \pi_k(a|\cdot)Q_j^k(\cdot,a)$ where

$$\pi_k(a|\cdot) = \text{SOFT-MAX}_\alpha^a((Q_r(\cdot,\cdot) + Y_k Q_g(\cdot,\cdot)).$$

Since $Q_j \in \mathcal{Q}_j$ for all $j$, and $0 \leq Y_k \leq \xi$, thus, $V_j \in \mathcal{V}_j$.

We now bound the $\epsilon$-covering number for the class of value function

**Lemma 13.** *There exists a* $\tilde{V}_j \in \mathcal{V}_j$ *parameterized by* $(\tilde{w}_r, \tilde{w}_g, \tilde{\beta}, \Lambda, \tilde{Y})$ *such that DIST* $(V_j, \tilde{V}_j) \leq \epsilon$ *where*

$$\text{DIST}(V_j, \tilde{V}_j) = \sup_x |V_j(x) - \tilde{V}_r(x)|. \tag{61}$$

Let $N_\epsilon^{V_j}$ be the $\epsilon$-covering number for the set $\mathcal{V}_j$, then,

$$\log N_\epsilon^{V_j} \leq d \log \left(1 + 8H \frac{\sqrt{dk}}{\sqrt{\lambda}\epsilon'}\right) + d^2 \log\left[1 + 8d^{1/2}\beta^2/(\lambda(\epsilon')^2)\right] + \log\left(1 + \frac{\xi}{\epsilon'}\right) \quad (62)$$

where $\epsilon' = \dfrac{\epsilon}{H2\alpha(1 + \xi + H) + 1}$

Note that $\epsilon$-covering number is dependent on $\xi$. This is because the policy depends on the Lagrange multiplier $Y$ which is upper bounded by $\xi$. Thus, we also need $\epsilon$-covering for the Lagrange multiplier in order to obtain $\epsilon$-close value function. Note that the $\epsilon$-covering does not depend on sample dependent terms. Rather it only depends on general $w_{j,h}$, $\Lambda$, and $Y$. Since the policy parameter is $\alpha$, we also have $\epsilon$-covering number is dependent on $\alpha$.

In order to prove the above lemma, we first state and prove some additional results.

We, first, obtain the $N_\epsilon^{Q_j}$ covering number for the set $\mathcal{Q}_j$. Towards this end, we first, introduce some notations.

**Definition 7.** *Let $\mathcal{C}_w^\epsilon$ be an $\epsilon/2$- cover of the set $\{w \in \mathbb{R}^d | \|w\| \leq 2H\sqrt{dk/\lambda}\}$ with respect to the 2-norm. Let $\mathcal{C}_\mathbf{A}^\epsilon$ be an $\epsilon^2/4$-cover of the set $\{\mathbf{A} \in \mathbb{R}^{d \times d} | \|\mathbf{A}\|_F \leq d^{1/2}\beta^2\lambda^{-1}\}$ with respect to the Frobenius norm.*

**Lemma 14.**

$$|\mathcal{C}_w^\epsilon| \leq (1 + 8H\sqrt{dk/\lambda}/\epsilon)^d, \quad |\mathcal{C}_\mathbf{A}^\epsilon| \leq [1 + 8d^{1/2}\beta^2/(\lambda\epsilon^2)]^{d^2} \quad (63)$$

*The $\epsilon$-covering number for the set $\mathcal{Q}_j$, for $j = r, g$, $N_\epsilon^{Q_j}$ of the set $\mathcal{Q}_j$ for $j = r, g$ satisfies the following*

$$\log N_\epsilon^{Q_j} \leq d \log\left(1 + \frac{8H\sqrt{dk}}{\sqrt{\lambda}\epsilon}\right) + d^2 \log[1 + 8d^{1/2}\beta^2/(\lambda\epsilon)^2] \quad (64)$$

*The distance metric is the $\infty$-norm, i.e., $\mathrm{dist}(Q_1, Q_2) = \sup_{x,a}|Q_1(x,a) - Q_2(x,a)|$.*

*Proof.* For notational simplicity, we represent $\mathbf{A} = \beta^2\Lambda^{-1}$, and reparamterized the class $\mathcal{Q}_j$ by $(w_j, \mathbf{A})$. Now,

$$\mathrm{dist}(Q_1, Q_2) = \sup_{x,a}|[w_1^T\phi(x,a) + \sqrt{\phi^T(x,a)\mathbf{A}_1\phi(x,a)}] - [w_2^T\phi(x,a) + \sqrt{\phi^T(x,a)\mathbf{A}_2\phi(x,a)}]|$$

$$\leq \sup_{\phi:\|\phi\|\leq 1}|[w_1^T\phi + \sqrt{\phi^T\mathbf{A}_1\phi}] - [w_2^T\phi + \sqrt{\phi^T\mathbf{A}_2\phi}]|$$

$$\leq \sup_{\phi:\|\phi\|\leq 1}|(w_1 - w_2)^T\phi| + \sup_{\phi:\|\phi\|\leq 1}\sqrt{|\phi^T(\mathbf{A}_1 - \mathbf{A}_2)\phi|}$$

$$= \|w_1 - w_2\| + \sqrt{\|\mathbf{A}_1 - \mathbf{A}_2\|} \leq \|w_1 - w_2\| + \sqrt{\|\mathbf{A}_1 - \mathbf{A}_2\|_F} \quad (65)$$

where the second-last inequality follows from the fact that $|\sqrt{x} - \sqrt{y}| \leq \sqrt{|x - y|}$. For matrices $\|\cdot\|$, and $\|\cdot\|_F$ denote matrix operator norm and the Frobenius norm respectively.

Recall that $\mathcal{C}_w$ is an $\epsilon/2$- cover of the set $\{w \in \mathbb{R}^d | \|w\| \leq 2H\sqrt{dk/\lambda}\}$ with respect to the 2-norm. Also recall that $\mathcal{C}_\mathbf{A}$ be an $\epsilon^2/4$-cover of the set $\{\mathbf{A} \in \mathbb{R}^{d \times d} | \|\mathbf{A}\|_F \leq d^{1/2}\beta^2\lambda^{-1}\}$. Thus, from Lemma 18,

$$|\mathcal{C}_w^\epsilon| \leq (1 + 8H\sqrt{dk/\lambda}/\epsilon)^d, \quad |\mathcal{C}_\mathbf{A}^\epsilon| \leq [1 + 8d^{1/2}\beta^2/(\lambda\epsilon^2)]^{d^2}$$

For any $Q_j \in \mathcal{Q}_j$, there exists a $\tilde{Q}_j$ parameterized by $(w_2, \mathbf{A}_2)$ where $w_2 \in \mathcal{C}_w^\epsilon$ and $\mathbf{A}_2 \in \mathcal{C}_\mathbf{A}^\epsilon$ such that $\mathrm{dist}(Q_j, \tilde{Q}_j) \leq \epsilon$. Hence, $N_\epsilon^{Q_j} \leq |\mathcal{C}_w^\epsilon||\mathcal{C}_\mathbf{A}^\epsilon|$, which gives the result since $\log(\cdot)$ is an increasing function. $\square$

Since the class of $Q$-function is independent of the policy we do not have $\xi$ and $\alpha$ in the $\epsilon$-covering number.

From the above lemma and since $Y_k \leq \xi$, we have the following,

**Corollary 1.** *If* $\mathrm{dist}(Q_r^k, \tilde{Q}_r) \leq \epsilon'$, $\mathrm{dist}(Q_g^k, \tilde{Q}_g) \leq \epsilon'$, *and* $|\tilde{Y}_k - Y_k| \leq \epsilon'$, *then,* $\mathrm{dist}(Q_r^k + Y_k Q_g^k, \tilde{Q}_r + \tilde{Y}_k \tilde{Q}_g) \leq \epsilon'(1 + \xi + H)$.

*Proof.* Note that $\tilde{Q}_j \in \mathcal{Q}_j$ belongs to the $\epsilon'$ covering of the set $\mathcal{Q}$.

$$
\begin{aligned}
\mathrm{dist}(Q_r^k + Y_k Q_g^k, \tilde{Q}_r + \tilde{Y}_k \tilde{Q}_g) &= \sup_{x,a} |(Q_r^k(x,a) + Y_k Q_g^k(x,a)) - (\tilde{Q}_r(x,a) + \tilde{Y}_k \tilde{Q}_g(x,a))| \\
&\leq \sup_{x,a} |(Q_r^k(x,a) + Y_k Q_g^k(x,a)) - (\tilde{Q}_r(x,a) + Y_k \tilde{Q}_g(x,a))| + \sup_{x,a} |(\tilde{Y}_k - Y_k) Q_g^k(x,a)| \\
&\leq \sup_{x,a} |Q_r^k(x,a) - \tilde{Q}_r(x,a)| + Y_k \sup_{x,a} |Q_g^k(x,a) - \tilde{Q}_g(x,a)| + \epsilon' H \\
&\leq \epsilon'(1 + Y_k) + \epsilon' H \\
&\leq \epsilon'(1 + H + \xi)
\end{aligned}
\tag{66}
$$

where the first inequality follows from the property of supremum and the norm. The second inequality follows from the norm, and the fact that $|\tilde{Y}_k - Y_k| \leq \epsilon'$, and $|Q_g^k(x,a)| \leq H$. The third inequality follows from the fact that $\mathrm{dist}(Q_j, \tilde{Q}_j) \leq \epsilon'$. $\qquad \square$

We now show that if the there exist $\tilde{Q}_j$, and $\tilde{Y}_k$ which are close to $Q_j$ and $Y_k$, then the soft-max policy is also close.

**Lemma 15.** *Suppose that $\pi$ is the soft-max policy (temp. coefficient $1/\alpha$) corresponding to the composite Q-functions $(Q_r^k + Y_k Q_g^k)$, i.e., $\forall a \in \mathcal{A}$*

$$
\pi(a|\cdot) = \text{SOFT-MAX}_\alpha^a((Q_r(\cdot,\cdot) + Y_k Q_g(\cdot,\cdot)).
$$

*$\tilde{\pi}$ is the soft-max policy vector with the same temp. coefficient $1/\alpha$ corresponding to the composite Q-function $(\tilde{Q}_r + \tilde{Y}_k \tilde{Q}_g)$, i.e, $\forall a \in \mathcal{A}$,*

$$
\tilde{\pi}(a|\cdot) = \text{SOFT-MAX}_\alpha^a((\tilde{Q}_r(\cdot,\cdot) + \tilde{Y}_k \tilde{Q}_g(\cdot,\cdot)).
$$

*then, for any state $x$,*

$$
||\pi(\cdot|x) - \tilde{\pi}(\cdot|x)||_1 \leq 2\alpha\epsilon'(1 + \xi + H) \tag{67}
$$

*where $\pi(\cdot|x) = \{\pi(a|x)\}_{a \in \mathcal{A}}$ and $\tilde{\pi}(\cdot|x) = \{\tilde{\pi}(a|x)\}_{a \in \mathcal{A}}$ when $\mathrm{dist}(Q_{j,h}^k, \tilde{Q}_j) \leq \epsilon'$ for $j = r, g$, and $|\tilde{Y}_k - Y_k| \leq \epsilon'$.*

*Proof.* Let $\text{Exp}^\alpha(P)$ be a soft-max corresponding to the vector $P$, i.e., the $i$-th component of $\text{Exp}^\alpha(P)$ is

$$
\frac{\exp(\alpha P_i)}{\sum_i \exp(\alpha P_i)}.
$$

Note from Theorem 4.4 in [29] then, we have

$$
||\text{Exp}^\alpha(P_1) - \text{Exp}^\alpha(P_2)||_1 \leq 2\alpha ||P_1 - P_2||_\infty \tag{68}
$$

for two vectors $P_1$ and $P_2$.

Now note that in our case for a given state $x$, $\pi$ is equivalent to $\text{Exp}^\alpha(Q_{r,h}^k(x,\cdot) + Y_k Q_{g,h}^k(x,\cdot))$, and $\tilde{\pi}$ is equivalent to $\text{Exp}^\alpha(\tilde{Q}_r(x,\cdot) + \tilde{Y}_k \tilde{Q}_g(x,\cdot))$. Then from (68) and the fact that $\mathrm{dist}(Q_{r,h}^k + Y_k Q_{g,h}^k, \tilde{Q}_r + \tilde{Y}_k \tilde{Q}_g) \leq \epsilon'(1 + \xi + H)$ (by Corollary 1) we have

$$
||\pi(\cdot|x) - \tilde{\pi}(\cdot|x)||_1 \leq 2\alpha\epsilon'(1 + \xi + H) \tag{69}
$$

Hence, the result follows. $\qquad \square$

Based on the above two lemmas we show that when the $Q$-functions are close, the value functions in the class $\mathcal{V}_j$ are also close.

**Lemma 16.** *There exists $\tilde{V}_j \in \mathcal{V}_j$ such that*

$$\text{DIST}(V_j^k, \widetilde{V}_j) \le H2\alpha\epsilon'(1 + \xi + H) + \epsilon', \tag{70}$$

*where* $\text{dist}(\tilde{Q}_j, Q_j) \le \epsilon'$, $\tilde{Q}_j \in \mathcal{Q}_j$ *for all* $j$;

$$\widetilde{V}_j(\cdot) = \sum_a [\tilde{\pi}(a|\cdot)\tilde{Q}_j(\cdot)],$$

$$\tilde{\pi}(a|\cdot) = \text{SOFT-MAX}_\alpha^a((\tilde{Q}_r(\cdot, \cdot) + \tilde{Y}_k \tilde{Q}_g(\cdot, \cdot)), \quad \forall a \in \mathcal{A}$$

$|\tilde{Y}_k - Y_k| \le \epsilon'$.

*Proof.* For any $x$,

$$
\begin{aligned}
&V_j^k(x) - \widetilde{V}_j(x) \\
&= |\sum_a \pi(a|x)Q_j^k(x, a) - \sum_a \tilde{\pi}(a|x)\tilde{Q}_j(x, a)| \\
&= |\sum_a \pi(a|x)Q_j^k(x, a) - \sum_a \pi(a|x)\tilde{Q}_j(x, a) + \sum_a \pi(a|x)\tilde{Q}_j(x, a) - \sum_a \tilde{\pi}(a|x)\tilde{Q}_j(x, a)| \\
&\le |\sum_a \pi(a|x)Q_j^k(x, a) - \sum_a \pi(a|x)\tilde{Q}_j(x, a)| + |\sum_a \pi(a|x)\tilde{Q}_j(x, a) - \sum_a \tilde{\pi}(a|x)\tilde{Q}_j(x, a)| \\
&\le \epsilon' + ||\pi(\cdot|x) - \tilde{\pi}(\cdot|x)||_1 ||\tilde{Q}_j(x)||_\infty \\
&\le \epsilon' + H2\alpha\epsilon'(1 + \xi + H)
\end{aligned}
\tag{71}
$$

where we use the fact that $\text{dist}(Q_j^k, \tilde{Q}_r) \le \epsilon'$, and $\sum_a \pi(a|x) = 1$ for the first term and the Holder's inequality in the second term for the second last inequality. For the last inequality, we use Lemma 15, and the fact that $\tilde{Q}_j(x, a) \le H$ for any $(x, a)$. Hence, we have the result. $\square$

Note that when $\alpha = \dfrac{\log(|\mathcal{A}|)K}{2(1 + \xi + H)}$ as we have in Algorithm 1, the right hand side in (70) becomes

$$\epsilon' + \log(|\mathcal{A}|)KH\epsilon' \tag{72}$$

We introduce one more notation which we use to prove Lemma 13.

**Definition 8.** *Let $\mathcal{C}_\xi^\epsilon$ be an $\epsilon$ cover for $Y \in [0, \xi]$. Hence, $|\mathcal{C}_\xi^\epsilon| \le \left(1 + \dfrac{\xi}{\epsilon}\right)$*

Note that $\mathcal{C}_\xi^\epsilon$ consists of points which is $\epsilon$-close to any point within the interval $[0, \xi]$. Since we have defined $\epsilon$-cover for all the parameters, we are now ready to prove Lemma 13.

*Proof.* Fix an $\epsilon$. Let $\epsilon' = \dfrac{\epsilon}{H2\alpha(1 + \xi + H) + 1}$, then from Lemma 16, we have $\text{DIST}(V_j^k, \widetilde{V}_j) \le \epsilon$. Thus, we only need to find parameters in the $\epsilon'$-covering of the $Q$-functions as described in Lemma 14 in order to obtain $\epsilon$-close value function.

Recall the Definition 7. Then, there exists $\tilde{w}_r, \tilde{w}_g \in \mathcal{C}_w^{\epsilon'}$ such that $||\tilde{w}_r - w_r|| \le \dfrac{\epsilon'}{2}$, $||\tilde{w}_g - w_g|| \le \dfrac{\epsilon'}{2}$.

Further, there exists $\mathbf{A}_2 \in \mathcal{C}_{\mathbf{A}}^{\epsilon'}$ such that $||\mathbf{A} - \tilde{\mathbf{A}}||_F \le \dfrac{\epsilon'^2}{4}$, $\mathbf{A} = \beta^2(\Lambda^k)^{-1}$, $\tilde{\mathbf{A}} = \beta^2(\tilde{\Lambda})^{-1}$, for some $\tilde{\Lambda}$, and $Y_k, \tilde{Y}_k$ such that $|Y_k - \tilde{Y}_k| \le \epsilon'$. Then we obtain $\tilde{Q}_j$ parameterized by $(\tilde{w}_j, \beta, \tilde{\Lambda})$ for $j = r, g$, such that $\text{dist}(Q_j, \tilde{Q}_j) \le \epsilon'$ (by Lemma 14).

Now define $\widetilde{V}_j = \sum_a \tilde{\pi}(a|\cdot)\tilde{Q}_j$, where

$$\tilde{\pi}(a|\cdot) = \text{SOFT-MAX}_\alpha^a((\tilde{Q}_r(\cdot, \cdot) + \tilde{Y}_k\tilde{Q}_g(\cdot, \cdot)).$$

Thus, from Lemma 16, we have $\text{DIST}(V_j^k, \tilde{V}_j) \le \epsilon$. Hence, there exists $\tilde{V}_j$ parameterized by $\tilde{w}_r, \tilde{w}_g, \tilde{Y}_k, \tilde{\mathbf{A}}$, such that $\text{Dist}(\tilde{V}_j, V_j^k) \le \epsilon$. Hence, $N_\epsilon^V \le |\mathcal{C}_w^{\epsilon'}||\mathcal{C}_{\mathbf{A}}^{\epsilon'}||\mathcal{C}_\xi^{\epsilon'}|$. Thus, from Lemma 14 and Definition 8, the $\epsilon$-covering number $N_\epsilon^{V_j}$ for the set $\mathcal{V}_j$ satisfies the following

$$\log N_\epsilon^{V_j} \le d \log \left( 1 + 8H \frac{\sqrt{dk}}{\sqrt{\lambda}\epsilon'} \right) + d^2 \log \left[ 1 + 8d^{1/2}\beta^2/(\lambda(\epsilon')^2) \right] + \log \left( 1 + \frac{\xi}{\epsilon'} \right).$$

Hence, the result follows. $\qquad\square$

From Lemma 13, note that we need $\epsilon'$ covering for the $Q$-functions where $\epsilon' = \dfrac{\epsilon}{(H2\alpha(1 + \xi + H) + 1)}$ if we need to bound $\text{DIST}(V_j, \tilde{V}_j)$ by $\epsilon$.

Now, we are ready to prove Lemma 8.

*Proof.* By Lemma 13, we know that there exists $\tilde{V}_j$ in the $\epsilon$-covering for $\mathcal{V}_j$ such that for every $x$,
$$V_j(x) = \tilde{V}_j(x) + \Delta V(x) \tag{73}$$
where $\sup_x \Delta V(x) \le \epsilon$.

Hence,

$$\left\| \sum_{\tau=1}^k \phi^\tau (V_j(x_\tau) - \mathbb{E}[V_j(x_\tau)|\mathcal{F}_{\tau-1}]) \right\|_{(\Lambda^k)^{-1}}^2 \le 2 \left\| \sum_{\tau=1}^k \phi^\tau (\tilde{V}_j(x_\tau) - \mathbb{E}[\tilde{V}_j(x_\tau)|\mathcal{F}_{\tau-1}]) \right\|_{(\Lambda^k)^{-1}}^2$$

$$+ 2 \left\| \sum_{\tau=1}^k \phi^\tau (\Delta V(x_\tau) - \mathbb{E}[\Delta V(x_\tau)|\mathcal{F}_{\tau-1}]) \right\|_{(\Lambda^k)^{-1}}^2 \tag{74}$$

The last expression is bounded by $\dfrac{8k^2\epsilon^2}{\lambda}$.

Now, we bound the first term. Note from Lemma 13 that in order to obtain $\tilde{V}_j$ which satisfies (73), we need to obtain we need $N_\epsilon^V$ number of elements to obtain such $(\tilde{w}_r, \tilde{w}_g, \beta, \tilde{\Lambda}, \tilde{Y})$. Such $\tilde{V}_j$ is independent of samples. Hence, we can use the Elliptical lemma for self-normalization (Theorem 2). From Theorem 2 and the union bound we obtain

$$\left\| \sum_{\tau=1}^k \phi^\tau (\tilde{V}_j(x_\tau) - \mathbb{E}[\tilde{V}_j(x_\tau)|\mathcal{F}_{\tau-1}]) \right\|_{(\Lambda^k)^{-1}}^2 \le 2H^2 \left[ d \log \left( \frac{k + \lambda}{\lambda} \right) + \log \left( \frac{N_\epsilon^V}{\delta} \right) \right] \tag{75}$$

where $N_\epsilon^V$ is upper bounded in (62). $\beta$ is equal to $C_1 dH\sqrt{\iota}$ for some constant $C_1$, and $\iota = \log(\log(|\mathcal{A}|)4dT/p)$. Further, $\xi = 2H/\gamma$ (by Definition 3). We obtain from (75)

$$\left\| \sum_{\tau=1}^k \phi^\tau (\tilde{V}_j(x_\tau) - \mathbb{E}[\tilde{V}_j(x_\tau)|\mathcal{F}_{\tau-1}]) \right\|_{(\Lambda^k)^{-1}}^2 \le$$

$$4H^2 \left[ \frac{d}{2} \log \left( \frac{k + \lambda}{\lambda} \right) + d \log \left( 1 + \frac{8H\sqrt{dk}}{\epsilon'\sqrt{\lambda}} \right) + d^2 \log \left( 1 + \frac{8d^{1/2}\beta^2}{\epsilon'^2\lambda} \right) + \log \left( 1 + \frac{2H}{\gamma\epsilon'} \right) + \log \left( \frac{4}{p} \right) \right] \tag{76}$$

where $\epsilon' = \dfrac{\epsilon}{(H2\alpha(1 + \xi + H) + 1)}$. Set $\epsilon = \dfrac{dH}{k}$, $\lambda = 1$. Thus, $\epsilon' = \dfrac{dH}{(2H\alpha(1 + \xi + H) + 1)k}$.

Plugging in the above, and putting $\alpha = \dfrac{\log(|\mathcal{A}|)K}{2(1 + \xi + H)}$, we obtain from (76)

$$\left\| \sum_{\tau=1}^k \phi^\tau (\tilde{V}_j(x_\tau) - \mathbb{E}[\tilde{V}_j(x_\tau)|\mathcal{F}_{\tau-1}]) \right\|_{\Lambda_k^{-1}}^2 \le C_2 H^2 d^2 \log \left( \frac{4(C_1 + 1)\log(|\mathcal{A}|)dT}{p} \right) \tag{77}$$

for some constant $C_2$ where $C_1 = 256(1 + 1/\gamma)$. Hence, the result follows.

$\square$

## F   Supporting Results

The following result is shown in [30] and in Lemma D.2 in [14].

**Lemma 17.** *Let $\{\phi_t\}_{t \geq 0}$ be a sequence in $\Re^d$ satisfying $\sup_{t \geq 0} ||\phi_t|| \leq 1$. For any $t \geq 0$, we define $\Lambda_t = \Lambda_0 + \sum_{j=0}^{t} \phi_j \phi_j^T \phi_j$. Then if the smallest eigen value of $\Lambda_0$ be at least 1, we have*

$$\log \left[ \frac{\det(\Lambda_h^{k+1})}{\det(\Lambda_h^1)} \right] \leq \sum_{k=1}^{K} (\phi_h^k)^T (\Lambda_h^k)^{-1} \phi_h^k \leq 2 \log \left[ \frac{\det(\Lambda_h^{k+1})}{\det(\Lambda_h^1)} \right] \tag{78}$$

**Theorem 2.** *[Concentration of Self-Normalized Process [30]] Let $\{\epsilon_t\}_{t=1}^{\infty}$ be a real-valued stochastic process with corresponding filtration $\{\mathcal{F}_t\}_{t=0}^{\infty}$. Let $\epsilon_t | \mathcal{F}_{t-1}$ be a zero mean and $\sigma$ sub-Gaussian, i.e., $\mathbb{E}[\epsilon_t | \mathcal{F}_{t-1}] = 0$, and*

$$\forall \zeta \in \Re, \quad \mathbb{E}[e^{\zeta \epsilon_t} | \mathcal{F}_{t-1}] \leq e^{\zeta^2 \sigma^2 / 2}. \tag{79}$$

*Let $\{\phi_t\}_{t=1}^{\infty}$ be a $\Re^d$-valued Stochastic process where $\phi_t \in \mathcal{F}_{t-1}$. Assume $\Lambda_0 \in \Re^{d \times d}$ is a positive-define matrix, let, $\Lambda_t = \Lambda_0 + \sum_{j=0}^{t} \phi_j \phi_j^T \phi_j$. Then for any $\delta > 0$ with probability at least $1 - \delta$, we have*

$$||\sum_{s=1}^{t} \phi_s \epsilon_s||_{\Lambda_t^{-1}}^2 \leq 2\sigma^2 \log \left[ \frac{\det(\Lambda_t)^{1/2} \det(\Lambda_0)^{-1/2}}{\delta} \right] \tag{80}$$

The next result characterizes the covering number of an Euclidean ball (Lemma 5.2 in [31]).

**Lemma 18.** *[Covering Number of Euclidean Ball] For any $\epsilon > 0$, the $\epsilon$-covering number of the Euclidean ball in $\mathbb{R}^d$ with radius $R$ is upper bounded by $(1 + 2R/\epsilon)^d$.*

We have used the following result from the optimization which is proved in Lemma 9 in [7].

**Lemma 19.** *Let $Y^*$ be the optimal dual variable, and $C \geq 2Y^*$, then, if*

$$V_{r,1}^{\pi^*}(x_1) - V_{r,1}^{\tilde{\pi}}(x_1) + C[b - V_{g,1}^{\tilde{\pi}}(x_1)]_+ \leq \delta \tag{81}$$

*then*

$$[b - V_{g,1}^{\tilde{\pi}}(x_1)]_+ \leq \frac{2\delta}{C}. \tag{82}$$

## G   Why does the Greedy Policy Fail?

In this section, using an example we show that the greedy-policy is not Lipschitz. Further, we can not use the greedy-policy on the composite $Q$-function to obtain the $\epsilon$-close covering for individual reward and utility value functions.

Consider the following toy-example: Suppose that the cardinality of the action space $|A|$ is 2. $Q_{r,h}(x, a_1) = M$, $Q_{r,h}(x, a_2) = 1$, $Q_{g,h}(x, a_1) = 1$, $Q_{g,h}(x, a_2) = M + \epsilon/2$. Consider $Y = 1$, then, greedy policy based on the composite $Q$-function is to choose action $a_2$.

Note the $\epsilon$-closest values can be anywhere in the ball within $\epsilon$ radius centered around $Q_{j,h}$, and $Y$. Assume that the closest $\epsilon$-cover for $Q_{j,h}$ be $\tilde{Q}_{j,h}$ such that $\tilde{Q}_{r,h}(x, a_1) = M + \epsilon/2$, $\tilde{Q}_{r,h}(x, a_2) = 1 - \epsilon/2$, $\tilde{Q}_{g,h}(x, a_1) = 1 + \epsilon/2$, $\tilde{Q}_{g,h}(x, a_2) = M$, and $\tilde{Y} = 1 - \epsilon/2$. Then, we have the composite $Q$-functions as

$$\tilde{Q}_{r,h}(x, a_1) + \tilde{Y} \tilde{Q}_{g,h}(x, a_1) = M + \epsilon/2 + (1 - \epsilon/2)(1 + \epsilon/2)$$
$$\tilde{Q}_{r,h}(x, a_2) + \tilde{Y} \tilde{Q}_{g,h}(x, a_2) = 1 - \epsilon/2 + (1 - \epsilon/2)M \tag{83}$$

Hence, it is clear that the greedy policy based on the composite $Q$-function is to choose action $a_1$. Hence, the policy is not Lipschitz. Thus, even though the change in the $Q$-function is only by $\epsilon$-amount the decision changes from taking action $a_2$ to taking action $a_1$ in a deterministic fashion.

Now, we see the changes in the value function. Since the policy is greedy and the policy is to choose $a_2$ for the $Q$-function $Q_{j,h}$, then, $V_{r,h}(x) = Q_{r,h}(x, a_2) = 1$. On the other hand, the policy is to choose $a_1$ for the $Q$-functions $\tilde{Q}_{j,h}$. Hence, $\tilde{V}_{r,h}(x) = M + \epsilon/2$, hence, $|V_{r,h}^k(x) - \tilde{V}_{r,h}(x)| > M - 1$, and can be made arbitrarily large by making $M$ arbitrarily large. Thus, the individual value functions can not be made close even though $Q_{j,h}$ and $\tilde{Q}_{j,h}$ are close. This is the reason we can not obtain $\epsilon$-covering number if the greedy policy is based on the composite $Q$-functions.

Note that in the unconstrained case (equivalent to $Y_k = 0$), the decision would be to choose $a_1$ for both $Q_{r,h}$ and $\tilde{Q}_{r,h}$ if the policy is set at the greedy one. Hence, the value function would only differ by at most $\epsilon$-amount. Hence, the greedy policy works for the unconstrained case.

## H Analysis for Zero Constraint Violation

The main idea behind attaining zero constraint violation is to consider the following tighter optimization problem–

$$\text{maximize } _{\pi \in \Delta(\mathcal{A}|\mathcal{S},\mathcal{H})} V_{r,1}^{\pi}(x_1) \quad \text{subject to } V_{g,1}^{\pi}(x_1) \geq b + \zeta. \tag{84}$$

Since we replace $b$ by $b + \zeta$, we are basically solving the above tighter optimization problem. By ensuring that $\zeta \leq \gamma/2$, we can ensure that Slater's condition is always satisfied, and strong duality holds. We show that by carefully choosing $\zeta$, we can achieve zero constraint violation with the same order on regret with respect to $T$. First, we introduce some notations which we use throughout this section.

Let $\pi^{\zeta,*}$ be the optimal solution of the optimization problem in (84). Since the Slater's condition holds, the strong duality holds by [19]. The optimal dual variable $Y^{\zeta}$ of this tighter problem is

$$Y^{\zeta} \leq \frac{V^{\pi^{\zeta,*}}(x_1) - V_{r,1}^{\bar{\pi}}(x_1)}{b + \gamma - (b + \zeta)} \leq 4H/\gamma \tag{85}$$

where the last inequality follows from the fact that $\zeta \leq \gamma/2$.

Now, we state the main result—

**Theorem 3.** *In Algorithm 1, replacing $b = b + \zeta$, and $\xi = 4H/\gamma$. We obtain, with probability $1 - p$,*

$$\text{Regret}(K) \leq C\mathcal{O}(\sqrt{d^3 H^3 T \iota^2}) + KH\zeta/\delta$$

$$\text{Violation}(K) \leq \max\{C'\mathcal{O}(\frac{2(1+\xi)}{\xi}\sqrt{d^3 H^3 T \iota^2}) - K\zeta, 0\} \tag{86}$$

*where $\zeta = \min\{C'\mathcal{O}\left(\frac{2(1+\xi)}{\xi}\frac{\sqrt{d^3 H^3 T \iota^2}}{K}\right), \gamma/2\}$.*

When $C'\mathcal{O}\left(\frac{2(1+\xi)}{\xi}\frac{\sqrt{d^3 H^3 T \iota^2}}{K}\right) \leq \gamma/2$, then the constraint is upper bounded by 0. Hence, for large enough $K$, we can achieve zero violation. However, by plugging the value of $\zeta$, we obtain the upper bound on regret as

$$\text{Regret}(K) \leq C\mathcal{O}(\sqrt{d^3 H^3 T \iota^2}) + C'H\mathcal{O}\left(\frac{2(1+\xi)}{\xi}\sqrt{d^3 H^3 T \iota^2}\right)$$

where we replace the upper bound of $\zeta$ by $C'\mathcal{O}\left(\frac{2(1+\xi)}{\xi}\frac{\sqrt{d^3 H^3 T \iota^2}}{K}\right)$. Thus, the upper bound on regret is $\tilde{\mathcal{O}}(\sqrt{d^3 H^5 T})$. Hence, the order on regret with respect to $T$ is maintained. However, there is an additional $H$ factor in front of the regret. A concurrent work [32] on model-based discounted horizon tabular setup using a generator model shows that extra $H$ is unavoidable if one wants to

achieve zero violation.* Even though the setup is different, it seems that the extra $H$ factor is unavoidable.

*Proof.* First, we prove the upper bound on regret. The regret can be decomposed as the following -

$$\text{Regret}(K) = \sum_{k=1}^{K}(V_{r,1}^{\pi^*}(x_1) - V^{\pi^{\zeta,*}}(x_1)) + \sum_{k=1}^{K}(V_{r,1}^{\pi^{\zeta,*}}(x_1) - V^{\pi_k}(x_1)) \tag{87}$$

The first term can be bounded with the help of the following lemma (the proof for finite state is in [13], the extension to linear MDP is provided after this proof)–

**Lemma 20.** *If $\pi^{\zeta,*}$ is the optimal solution of (84), then*

$$V_{r,1}^{\pi^*}(x_1) - V_{r,1}^{\pi^{\zeta,*}}(x_1) \le H\frac{\zeta}{\gamma}. \tag{88}$$

Since the tighter optimization problem is also CMDP, we note that the second term in the right hand side of (87) is essentially the regret of the tighter CMDP.

Hence, from Theorem 1 and Lemma 20 we obtain the expression of the regret bound in Theorem 3.

**Constraint Violation**: Again applying Theorem 1 to the tighter optimization problem (84), we obtain

$$\sum_{k=1}^{K}(b + \zeta - V_{g,1}^{\pi_k}(x_1))_+ \le C'\mathcal{O}(\frac{2(1+\xi)}{\xi}\sqrt{d^3H^3T\iota^2})$$

$$\sum_{k=1}^{K}(b + \zeta - V_{g,1}^{\pi_k}(x_1)) \le \sum_{k=1}^{K}(b + \zeta - V_{g,1}^{\pi_k}(x_1))_+ \le C'\mathcal{O}(\frac{2(1+\xi)}{\xi}\sqrt{d^3H^3T\iota^2})$$

Hence,

$$\sum_{k=1}^{K}(b + \zeta - V_{g,1}^{\pi_k}(x_1)) \le C'\mathcal{O}(\frac{2(1+\xi)}{\xi}\sqrt{d^3H^3T\iota^2})$$

Thus,

$$\sum_{k=1}^{K}(b - V_{g,1}^{\pi_k}(x_1)) \le C'\mathcal{O}(\frac{2(1+\xi)}{\xi}\sqrt{d^3H^3T\iota^2}) - K\zeta$$

Hence, we have

$$\sum_{k=1}^{K}[b - V_{g,1}^{\pi_k}(x_1)]_+ \le \max\{C'\mathcal{O}(\frac{2(1+\xi)}{\xi}\sqrt{d^3H^3T\iota^2}) - K\zeta, 0\}$$

Thus, the result follows. □

*Proof of Lemma 20*: We, first, introduce a few notations.

Let $\nu_h^{\pi}(x)$ for $h = 2, \ldots, H$ be

$$\nu_h^{\pi}(x) = \int_{x'}\sum_a \pi_h(a|x')\phi(x', a)^T\mu_{h-1}(x)d\nu_{h-1}^{\pi}(x')$$

and $\nu_1(x)$ is the initial distribution of the state. $\nu_h^{\pi}(x)$ is the distribution of the state at step $h$ while following the policy $\pi$. It is the state occupation measure at step $h$.

Also, $\nu_h(x, a) = \pi_h(a|x)\nu_h(x)$ is the state-action occupation measure at step $h$. Hence,

$$V_{j,1}^{\pi}(x_1) = \sum_h \int_{x,a} j_h(x, a)d\nu_h(x, a) \tag{89}$$

---

*[32] provides a sample complexity guarantee for the discounted horizon setup with discount factor $\gamma$. One can convert the result in the discounted setup to the episodic setup by equating $1/(1 - \gamma) = H$.

Now, $\nu_h^*(x, a)$ corresponds to the state-action occupancy measure for the optimal policy $\pi^*$. Then, $\nu_h^\zeta(x, a) = (1 - \zeta/\gamma)\nu_h^*(x, a) + \zeta/\gamma\nu_h^{\bar{\pi}}(x, a)$. Now, we are resy to prove Lemma 20.

We have

$$\sum_h \int_{x,a} g_h(x, a)d\nu_h^\zeta(x, a) \geq (1 - \zeta/\gamma)b + \zeta/\gamma(b + \gamma) = b + \zeta \tag{90}$$

Hence, the state-action occupancy measure $\nu_1^\zeta(x, a)$ is feasible for the tightened CMDP. Now, we have

$$\sum_h \int_{x,a} r_h(x, a)d\nu_h^\zeta(x, a) = (1 - \zeta/\delta) \sum_h \int_{x,a} r_h(x, a)d\nu_h^*(x, a) + \zeta/\delta \sum_h \int_{x,a} r_h(x, a)d\nu_h^{\bar{\pi}}(x, a)$$
$$\geq (1 - \zeta/\delta)V_{r,1}^*(x_1)$$

Since $\nu_1^\zeta(x, a)$ is feasible, then $V_{r,1}^{\pi^{\zeta,*}}(x_1) \geq \sum_h \int_{x,a} r_h(x, a)d\nu_h^\zeta(x, a)$. Thus,

$$V_{r,1}^*(x_1) - V_{r,1}^{\zeta,*}(x_1) \leq \zeta/\delta V_{r,1}^*(x_1) \leq \frac{\zeta}{\delta}H \tag{91}$$

Hence, the result follows. $\qquad\square$

**Remark 3.** *Note that [13] and [6] use Lyapunov Drift analysis to obtain constraint violation for finite state space (tabular setting). To obtain the zero violation, their approach is also similar to ours where they also consider a tighter optimization problem, and then, carefully choosing the parameter of the tighter optimization. However, one key difference is that– they did not use any upper bound on the dual variable. Rather they rely on the Hajek's Lemma [33] to establish a finite bound on the dual variable for the good event. The question is whether we can use similar technique in the model-free linear function approximation.*

*Note that we need to have an upper bound on the dual-variable (irrespective of the good and bad event) since the $\epsilon$-covering number (Lemma 13) depends on $\xi$, the upper bound on the dual variable. Hence, we need to truncate the dual variable if it exceeds $\xi$. However, if we truncate the dual-variable, we can not use the Lyapunov-Drift analysis to bound the violation. Since in the Lyapunov-drift analysis, it relies on the fact that the magnitude of the dual-variable (or, the queue length) lower bounds the total violation (see the analysis at page 21 in [6]). Since the magnitude of the dual-variable is bounded for the good-event, one then obtain the upper bound of the violation. However, one can not extend the same analysis if we truncate the dual-variable.*

*Hence, our analysis is based on the results from the convex optimization (Lemma 19). Our approach to obtain zero violation for large enough $K$ relies on different tools compared to the Lyapunov-Drift analysis and new of a kind.*

# I Comparison with other approaches to show uniform concentration lemma for individual value function in model-free setup

[34] considered a zero-sum. linear Markov game setup. The paper proposed an approach where they truncate $w_h^k$, $\Lambda_h^k$ to $\epsilon$-close value of $w$, and $\Lambda$ respectively. Subsequently, they obtained equilibrium policies using these $\epsilon$-close values. Then the proposed algorithm uses the above equilibrium policy attained using $\epsilon$-close values. Since these $\epsilon$-close values are predetermined using the $\epsilon$-covering set of $w$ and $\Lambda$ (as we have described the $\epsilon$-covering set in Lemma 14), hence, one can apply uniform concentration lemma for each individual value function [34] with error of at most $\epsilon$. We can also apply the similar trick in our set-up where we truncate the obtained $w_{j,h}^k$, and $\Lambda_h^k$ to one of the $\epsilon$-close values and then we can set the policy as the greedy one with respect to the composite approximated state-action value function. The above method would also provide a log $\epsilon$-covering number of $\mathcal{O}(\log(K))$ for each individual value function.

However, our soft-max based approach has several advantages compared to the above approach.

(i) *Computation efficiency.* The alternative method explicitly rounds the $Q$-function to its $\epsilon$-close one *in the algorithm*, which requires an additional $O(d^2)$ computation even with an efficient implementation (i.e., rounding on the fly) (See Section 3.3 in [34]). In contrast, our soft-max works directly

with the actual $Q$-function without any $\epsilon$-close rounding. That is, $\epsilon$-net argument is only used *in the analysis* in our setting.

(ii) *Stochastic policy.* A key fact about constrained MDPs is that the optimal policy is usually stochastic. Although the greedy-policy in the alternative method can approach the optimal policy in an average sense, in each episode, it could be far away from the optimal policy. Even though in our setting, $\alpha$ scales with $K$, our approach puts 'almost' similar probability among the composite $Q$-functions with 'almost' same values. However, in the greedy policy, it chooses the action corresponding to the highest state-action value function. Thus, such an approach can never be close to the optimal policy.

iii) *General applicability.* The alternative method relies heavily on the fact that there exists an efficient implementation of rounding on the fly in linear MDPs so that there is no need to construct an explicit $\epsilon$-net. However, beyond linear MDPs, it might not be possible to find an efficient rounding algorithm on the fly, and hence an even larger additional computation is required. In contrast, our soft-max based algorithm builds on the intrinsic smoothness-approximation trade-off in soft-max to establish uniform concentration and approximate optimism, which could potentially be generalized to other settings. Further, soft-max policy is popular, hence, our approach would provide the base for proving regret and violation bound for the general function approximation setup beyond the linear function approximation.