# OpenReview forum: "Provably Efficient Model-Free Constrained RL with Linear Function Approximation"
_NeurIPS.cc/2022/Conference — NeurIPS 2022 Accept_

### Official Review · Reviewer_4Xoe · 2022-07-02

**Rating:** 7
**Confidence:** 3
**Soundness:** 3 good
**Presentation:** 3 good
**Contribution:** 3 good

**Summary:**

The paper considers the episodic, model-free, simulator-free, constrained Markov Decision Process (CMDP) framework with a linear function approximator.
The main contribution is the primal-dual algorithm that is based on the LSVI-UCB algorithm. The paper also provides the regret and constraint violations' bounds that do not depend on the state space cardinality. Instead, the bounds depend on on the dimension of feature mapping $d$. In addition, the presented algorithm can achieve a zero constraints' violation given a sufficiently large number of episodes $K$ as proven in theorem 3.
The paper also compares the presented algorithm with other algorithms that consider the model-free CMDP case, and mentions the shortcomings of them (e.g. their need for a simulator, being model-based).

**Questions:**

**Question 1:** With linear function approximators, finding a good performing set of features significantly affects the approximations' accuracy. Can the author(s) discuss how a set of features $\phi(s,a)$ can be chosen for a linear approximator?

I am willing to increase my score if the author(s) address the weaknesses I mentioned as well as the question I have here.

**Limitations:**

**Limitations:** work only considers a linear function approximator. It would be nice to expand this approach to a quadratic function approximator or a kernel-basis functions. The author(s) also mention the possibility of finding tighter bounds given *d* and *H*, and I am pleased that they addressed this bound gap.


**Societal impact:** Not applicable to this work.

**Strengths And Weaknesses:**

**Strengths:**

The primal-dual extension to the LSVI-UCB algorithm is non-trivial and is a good contribution to the CMDP literature. The author(s) also clearly presented how their algorithm differentiates from other model-free or tabular algorithms as shown in table 1. The constraints' violations was also shown to go to zero asymptotically w.r.t. the number of episodes.

In addition, theorem 3 (and its proof in the appendix) is a nice addition to the finite-time regret\constraint violation analysis. I read the proofs provided in the appendix, and to the best of my knowledge they appear to be correct in their analysis.


**Weaknesses:**
The paper's biggest weakness is its lack of experimental evaluation of the proposed algorithm to support the theoretical bounds. Some of the cited work (Triple-Q algorithm) do present results for the tabular and function approximation cases.

In addition, the Triple-Q algorithm (cited as [13] inn the paper) was also extended to the neural networks' function approximation case. This raises the point if the proposed primal-dual algorithm should be extended to other function approximators (possibly radial-basis function and not neural networks). The claim on lines 86 and 87 is hence inaccurate since other work can already handle continuous state spaces.

Minor comments I have:

Line 45 *value function or a policy*: value function of a policy.

Line 87 *regret bound shown in the above paper*: it's better if the author(s) cites the paper again to avoid confusion.

The Gram matrix $\Lambda_h^k$ is first mentioned on line 196 but it's referenced earlier in Algorithm 1 and its description. The symbol definition should be introduced earlier in the paper.

Line 354: *Whether we can the tighten the*: Whether we can tighten the.

Line 514 *we, now, recursively*: we now, recursively.


--------------------------------------
### After rebuttal:

I read the responses by the author(s) and also the revised manuscript. I am pleased to see that the author(s) implemented an experiment in less than a week. I also find their response regarding picking a good feature set $\phi(s,a)$ sufficient.

I increased my score from 6 to 7.

---

> ### Author Response · Authors · 2022-08-02
> **Part 1 of Response to Reviewer 4Xoe**
>
> We appreciate your time and thoughtful evaluation of our paper. We recap your comment and present our detailed response as follows. Several of your questions raise challenging open problems that we have thought about, and which we believe might constitute interesting research directions with potentially great practical significance. We would also be happy to provide further clarifications if suitable.
>
> > *The paper's biggest weakness is its lack of experimental evaluation of the proposed algorithm to support the theoretical bounds… In addition, the Triple-Q algorithm (cited as [13] in the paper) was also extended to the neural networks' function approximation case. This raises the point if the proposed primal-dual algorithm should be extended to other function approximators (possibly radial-basis function and not neural networks). The claim on lines 86 and 87 is hence inaccurate since other work can already handle continuous state spaces.*
>
> - We have now conducted an experimental study that we were able to undertake in the last week. We present the result in Appendix J (page 29). In the experiments, we observe that the regret scales sub-linearly with $K$ (Figure 1, page 30). Thus, it corroborates our theoretical findings. We also observe that the constraint violation becomes close to $0$ as $K$ grows. The violation oscillates since our dual variable also oscillates as it adaptively changes to control how it weighs the utility value function with respect to the reward value function. We will add more experimental results in the final paper.
>
> - The reviewer is correct to point out that triple-Q indeed does experimental evaluations on neural networks and tabular set-up. **However, triple-Q does not provide any theoretical guarantees beyond the tabular setup.** We will, thus, modify the statement in lines 86 and 87 as the following– there is no work with a provable guarantee for the continuous state-space in the constrained MDP space. We contribute in this space. As a by-product, we also improve the bound achieved by the triple-Q paper.
>
> - In the numerical evaluation of triple-Q, relies on obtaining an estimate of the $Q$-functions using actor-critic setup. We can envision that using a similar approach, our algorithm can be extended to perform numerical evaluations on continuous state-space with non-linear function approximation. For example, using a neural network, we can find a good feature-space representation. In the final manuscript, we will add more numerical evaluations along this line.
>
> - In the following, we provide some ideas on how our approach can be extended to achieve theoretical bounds for other function approximation setups. Our algorithm is an adaptation of the LSVI-UCB in the primal-dual domain where we replace the greedy policy with the soft-max policy in order to obtain  $\epsilon$-covering number for individual value function set polynomial in $K$. Whether we can extend our approach to non-linear function approximation certainly constitutes an important research question. Towards this end, we want to point out that, recently, algorithms have been proposed for unconstrained low Bellman-eluder rank functions [R1]. [R1] shows that linear kernel MDP has a low Bellman-eluder dimension (Appendix B). The algorithm proposed in [R1] also relies on showing that $\epsilon$-covering number for the value function set scales polynomially with $K$. Hence, we conjecture that with a similar approach, our algorithm can also be combined with the algorithm proposed in [R1] to extend the approach for the CMDP for low Bellman-Eluder Rank MDPs. Complete characterization constitutes an important future research direction.
>
> [R1]. Jin, Chi, Qinghua Liu, and Sobhan Miryoosefi. "Bellman eluder dimension: New rich classes of rl problems, and sample-efficient algorithms." Advances in neural information processing systems 34 (2021): 13406-13418.

---

> > ### Author Response · Authors · 2022-08-02
> > **Part 2 of Response to Reviewer 4Xoe**
> >
> > In this part, we respond to your comment on finding good features $\phi(x,a)$.
> >
> > > *With linear function approximators, finding a good performing set of features significantly affects the approximations' accuracy. Can the author(s) discuss how a set of features
> > $\phi(s,a)$ can be chosen for a linear approximator?*
> >
> > - Note that feature space learning is an active area of research [R1-R4] for linear approximation setup. This class of work is classified as representation learning. The most promising technique to determine good feature space vector $\phi(\cdot,\cdot)$ is to use the maximum likelihood estimation (as described in Definition 3 in [R2]). We again divide the literature in model-based and model-free domain. [R2] seeks to fit $\phi$, and $\mu$ to estimate the transition probability kernel using MLE. Hence, they consider a model-based approach. In [R1] and [R4], MLE is used to fit $w$, and $\phi$ to estimate the $Q$-function directly by solving an optimization problem where both $w$, and $\phi$ are decision variables (hence, it is a model-free approach). The $Q$-function candidate is chosen from a class of functions (see equation (1) in [R1]). Note that neural networks can be used to approximate those optimal solutions. We will include them in the final manuscript.
> >
> > - Using a similar technique, we can try to learn a good feature representation $\phi(x,a)$.  However, such a characterization is beyond the scope of this paper.
> >
> >
> >  > *Regarding the minor comments.*
> >
> > - We thank you for pointing those out. We have rectified the typos (plese see line number 45, line number 357, line number 514) and added the reference ( line number 87). We have also introduced $\Lambda_h^k$ before Algorithm 1. The changes are marked in blue.
> >
> >
> >
> > [R1]. Zhang, Xuezhou, Yuda Song, Masatoshi Uehara, Mengdi Wang, Alekh Agarwal, and Wen Sun. "Efficient reinforcement learning in block mdps: A model-free representation learning approach." In International Conference on Machine Learning, pp. 26517-26547. PMLR, 2022.
> >
> > [R2]. Agarwal, Alekh, Sham Kakade, Akshay Krishnamurthy, and Wen Sun. "Flambe: Structural complexity and representation learning of low rank mdps." Advances in neural information processing systems 33 (2020): 20095-20107.
> >
> > [R3]. Ren, Tongzheng, Tianjun Zhang, Csaba Szepesvári, and Bo Dai. "A free lunch from the noise: Provable and practical exploration for representation learning." arXiv preprint arXiv:2111.11485 (2021).
> >
> > [R4]. Modi, Aditya, Jinglin Chen, Akshay Krishnamurthy, Nan Jiang, and Alekh Agarwal. "Model-free representation learning and exploration in low-rank mdps." arXiv preprint arXiv:2102.07035 (2021).

---

### Official Review · Reviewer_EWtE · 2022-07-10

**Rating:** 7
**Confidence:** 5
**Soundness:** 4 excellent
**Presentation:** 3 good
**Contribution:** 3 good

**Summary:**

This paper studied the episodic CMDPs with linear function approximation and proposed the LSVI-UCB based algorithm which can achieve $O(\sqrt{T})$ regret and $O(\sqrt{T})$ constraint violation bounds.  The author also showed that zero constraint violation is achievable for a large $T.$

**Questions:**

1. This paper indicates that their algorithm is model-free in the sense that the algorithm does not need to estimate the transition kernels. My concerns come from the comparison between this paper and [7]. Although Linear MDP and linear kernel MDP have different assumptions on the model, your algorithms both need to solve a LSTD problem at each episode. [7] needs to estimate the transition kernel in the tabular setting because of the assumption from the linear kernel assumption. I am wondering if your algorithm needs to estimate the transition kernels in the tabular setting?
2. The LSTD-based algorithms are well studied in both bandit and RL settings. Can the authors clarify what the novel technical contributions of this paper are?

**Limitations:**

I don't think this paper has any potential negative societal impact.

**Strengths And Weaknesses:**

**Strengths**
1. The paper is well-written and well-organized.
2. The paper proposes a model-free, simulator-free reinforcement learning algorithm for CMDPs with linear function approximation that can achieve $\sqrt{T}$ regret and zero constraint violation.
3. The paper proves that zero constraint violation is possible when $T$ is sufficiently large.

**Weaknesses**
1. The memory complexity is high, since the algorithm needs to store all the historical states and actions.

---

> ### Author Response · Authors · 2022-08-02
> **Part 1 of Response to Reviewer EWtE**
>
> We appreciate your time and thoughtful evaluation of our paper. We recap your comment and present our detailed response as follows. We would also be happy to provide further clarifications if suitable.
>
>  >*The memory complexity is high, since the algorithm needs to store all the historical states and actions.*
>
> - We would like to point out that $w_{r,h}^k$, $w_{g,h}^k$ and $\Lambda_h^k$ can be updated recursively. We only need to store the states and actions encountered in the trajectories. Thus, the memory complexity in our algorithm is of the same order as the unconstrained version (LSVI-UCB). We would also like to point out that [7] (which proposes a model-based algorithm for CMDP with linear function approximation) also needs to store all the historical states, actions, and even the estimated value functions for the encountered states. Characterizing algorithms with provably optimal regret and violation bound with limited memory is certainly an interesting future research direction. We will explicitly mention that in the conclusion.
>
> > *[..] My concerns come from the comparison between this paper and [7]... [7] needs to estimate the transition kernel in the tabular setting…. I am wondering if your algorithm needs to estimate the transition kernels in the tabular setting?*
>
> -  Thank you for raising this point. In the tabular set-up, we **do not need to estimate the transition kernel** unlike in [7] (Please see eqn. (9) in [7]) . Basically, we can revert back to the tabular case by setting $\phi(s,a)=e_{s,a}$ where $e_{s,a}$ is a $d$-dimensional (here $d=|\mathcal{S}||\mathcal{A}|$) vector where $e_{s,a}=1$ for state-action pair $(s,a)$ and zero for other values of state and action. The $w_{r,h}$ vector update becomes as the following $w_{r,h}^k(x,a)=\dfrac{1}{n_h^k(x,a)+\lambda)}\sum_{\tau=1}^{n_h^k(x,a)}(r_h(x_h^{\tau},a_h^{\tau})+V_{r,h+1}^k(x_{h+1}^{\tau}))$ where $n_h^k(x,a)$ is the number of times the state-action pair $(x,a)$ has been encountered at step $h$ till episode $k$.
>
> - The $Q_{r,h}^k$ update will be $Q_{r,h}^k(x,a)=\min${$\langle w_{r,h}^k(x,a),\phi(x,a)\rangle +\beta \sqrt{1/(n_h^k(x,a)+\lambda)},H$}.  In a similar way, we can update $Q_{g,h}^k$. Hence, we do not need to estimate any transition probability kernel. Using these values, we obtain our policy based on the soft-max function.
>
> We will include the above in the appendix of the final manuscript.

---

> > ### Author Response · Authors · 2022-08-02
> > **Part 2 of Response to Reviewer EWtE**
> >
> > Here, we respond to your question regarding the technical novelty of our paper.
> >
> > >*The LSTD-based algorithms are well studied in both bandit and RL settings. Can the authors clarify what the novel technical contributions of this paper are?*
> >
> > - The reviewer is correct to point out that LSTD-based algorithms are well studied in both bandit and RL settings. However, the main difference is that for the model-free RL algorithm we need to show value-aware uniform concentration bound for the value function set (See Section 4 in [14]). The basic idea is to control the fluctuations in the least square value iteration for each individual value function (See step 2 in Section 4.2 of our paper, page 7) by showing that the expression in Line 281 is upper bounded by $d\mathcal{O}(\log(K))$. However, we cannot use the self-normalized concentration result because of the dependence of $V_{h+1}^k$ on the samples {$x_{h+1}^k$}$_{\tau=1}^{K}$.
> >
> > - We need to rely on the uniform concentration bound on the value function class as has been done in the unconstrained case [14] (see Section 4 there). Specifically, we need to show that each individual value function class has log  $\epsilon$-covering number which scales with $d\log(K)$ (see Lemma 13).  **However, showing the above is  significantly more challenging  than the unconstrained setup.** The issue is that when there is no constraint, one can show that the class of $Q$-function has log $\epsilon$-covering number which  scales with $\log(K)$ (see Lemma D.6 in [14]). Now, using the contraction property of the max operator, one can show that the value function class also has log $\epsilon$-covering number which is upper bounded by $\log(K)$. However, in the constrained case, the decision is taken with respect to the composite state-action value function. Hence, even though the individual $Q$-function class has a log $\epsilon$-covering number, which is upper bounded by $\log(K)$ (see Lemma 14, page 22), the greedy policy with respect to the composite $Q$-functions is unable to provide such a bound for the individual value function as it is not Lipschitz (see Appendix G for an example, page 26). We introduce the soft-max policy since it is Lipschitz. Since individual $Q$-functions have log $\epsilon$-covering number which is upper bounded by $\log(K)$, using the Lipschitz property of the soft-max, we can also show that individual value function also has log $\epsilon$-covering number which is upper bounded by $\log(K)$ (Lemma 13, page 21) by controlling the parameter $\alpha$ of the Soft-max. **This is main novel technical contribution of our paper.**
> >
> > - Note that since the policy is not greedy, we cannot obtain optimism result with respect to the combined value function. However, again, using the property of $\alpha$ we show that we can bound the optimism term ($\mathcal{T}_1$ in Lemma 2, page 7) by a constant using the property of the soft-max (Lemma 11, line 550 ). Please see Section 4.2 for a detailed analysis.

---

> > > ### Author Response · Authors · 2022-08-08
> > > **Follow up**
> > >
> > > Since the discussion period is closing soon, we just wanted to check in and ask if the rebuttal clarified and answered the questions raised in your review. We would be very happy to engage further if there are additional questions!

---

### Official Review · Reviewer_F93d · 2022-07-13

**Rating:** 5
**Confidence:** 3
**Soundness:** 2 fair
**Presentation:** 3 good
**Contribution:** 2 fair

**Summary:**

The paper studies episodic constrained MDP problems. The authors propose a model-free value-based RL algorithm with linear function approximation and establish sublinear regret and violation.

**Questions:**

- What is $\xi$ in your analysis? Any dependence on $H$?

- It is useful to discuss some work in linear setting: A Simple Reward-free Approach to Constrained Reinforcement Learning

- The proof outline is helpful, but it is useful to highlight the new key techniques. If I set $Y=0$, would this become the standard LSVI-UCB and analysis carries over?


**Ethics Review Area:**

["I don’t know"]

**Limitations:**

Yes.

**Strengths And Weaknesses:**

## Originality

- The proposed primal-dual algorithm is new and it works for constrained MDPs with linear function approximation, which is an important constrained RL setting.

- The proof has some novel ingredients, e.g., uniform concentration of value functions.

## Quality & Clarity

- The paper is well written and all claims are justified via proofs. The authors also discussed improvements and limitations in the end of the paper.

## Significance

- The proposed primal-dual algorithm is based LSVI-UCB while taking account for constraints in the linear function approximation setting. This is a new advance in learning constrained MDPs with large state spaces.

- The analysis generalizes the uniform concentration bound in LSVI-UCB to the constrained case using an intermediate softmax policy update.

- This regret and violation bounds are near-optimal. They improve the dependence on $H$ compared with the existing methods, while $d$-dependence is slightly worse.

---

> ### Author Response · Authors · 2022-08-02
> **Response to Reviewer F93d**
>
> We appreciate your time and thoughtful evaluation of our paper. We recap your comment and present our detailed response as follows. We would also be happy to provide further clarifications if it is suitable.
>
>
> > *What is  $\xi$  in your analysis? Any dependence on H?*
>
> - In Definition 3 (Line 171, page 5), we have characterized the value of $\xi$. The value of $\xi=2H/\gamma$. It is linearly dependent on $H$. Note that $\xi$ is twice the value of the upper bound of the dual variable (Lemma 1). Using $\xi$ to truncate the dual variable is a common practice in the primal-dual type algorithm for the CMDP setup.  Please see the references [1] (Assumption 2), [7] (Assumption 1, Line 1 in Algorithm 1). In our regret and violation, we replace $\xi$ with the value of $2H/\gamma$ in order to compare with the state-of-the-art results (Table 1, page 3).
>
> > *It is useful to discuss some work in linear setting: A Simple Reward-free Approach to Constrained Reinforcement Learning*
>
> - The set-up considered in the simple reward-free approach to constrained reinforcement learning [R1] is quite different from ours. In [R1], the goal is to obtain a sample complexity guarantee. However, in our approach, we consider the regret and violation guarantee. Hence, we provide *any time* performance guarantee. Thus, the algorithms are inherently different. Note that using the explore-then commit algorithm proposed in [R2], it is possible to achieve a regret of $\tilde{\mathcal{O}}(d^3H^6T^{2/3})$ for $\tilde{\mathcal{O}}(d^3H^6/\epsilon^2)$ sample-complexity guarantee achieved in [R1]. **Hence, the regret bound provided by [R1] would be sub-optimal both in terms of $T$ and $H$ compared to ours**. **Further,  we show that we can achieve zero violation for large enough $K$ while maintaining the same order on regret.** We will include the above comparison with [R1] in the final manuscript.
>
> - We now explain why our algorithm outperforms the algorithm proposed in [R1]. The algorithm proposed in [R1] consists of two phases-- in the exploration phase,  it employs an exploration algorithm to select actions in order to reduce the confidence interval. Subsequently, in the planning phase, the algorithm in [R1] computes the policy which maximizes the composite estimated state-action value function of reward and the utility (constraint). The dual variable is updated in a similar manner as ours depending on the estimated value function for utility. Note that high regret and violation can be incurred in the exploration phase as the goal is different in that phase. In our approach, we do not divide into the exploration and planning phase. Hence, we achieve a better performance guarantee.
>
> > *The proof outline is helpful, but it is useful to highlight the new key techniques. If I set $Y=0$, would this become the standard LSVI-UCB and analysis carries over?*
>
> - First, we would like to point out the difference between our algorithm and LSVI-UCB. Our algorithm is based on the soft-max policy unlike the greedy policy in LSVI-UCB. Specifically, in our algorithm, at a given state the action is chosen by applying the soft-max on the combined state-action value function of reward and the scaled state-action value function for utility.
>
> - Now, we point out why we have to introduce soft-max policy (please see Section 4.2, step 2 (now marked in blue) page 8,  line 298). The main reason behind introducing the soft-max policy is that for a model-free algorithm, one needs to show a uniform concentration bound over the class of individual value functions. Specifically, one needs to show that $\epsilon$-covering number for each individual value function set only grows polynomially with $K$ even though the decision is taken based on the combined value function. The greedy policy is unable to do that as it is not Lipschitz continuous (please see Appendix G for example). Being a Lipschitz continuous, soft-max can provide such a small $\epsilon$-covering number.
>
> - As a by-product, even when $Y=0$ (dual variable is $0$), our algorithm will still be different compared to LSVI-UCB. Hence, the algorithm is fundamentally different and so is the analysis. However, we would like to point out that the order of regret will be the same as in LSVI-UCB in terms of $d$, $H$, and $K$ when $Y=0$.  We will also include the above discussion in the final manuscript. Note that when Y=0, the policy is based only on the reward state-action value function. Hence, naturally, the greedy-policy can provide an $\epsilon$-covering number which only grows polynomially with $K$ (please see the discussion on Step 2 in Section 4.2).
>
> [R1]. Miryoosefi, Sobhan, and Chi Jin. "A simple reward-free approach to constrained reinforcement learning." International Conference on Machine Learning. PMLR, 2022.
>
> [R2]. Jin, Chi, Zeyuan Allen-Zhu, Sebastien Bubeck, and Michael I. Jordan. "Is Q-learning provably efficient?." Advances in neural information processing systems 31 (2018).

---

### Official Review · Reviewer_K6Mq · 2022-07-21

**Rating:** 7
**Confidence:** 2
**Soundness:** 3 good
**Presentation:** 2 fair
**Contribution:** 3 good

**Summary:**

The paper deals with providing an algorithmic scheme which is provable efficient for Model-free RL in a setting where there are additional constraints on the learned policy in the sense that it also needs to ensure that the expected utility is above a given threshold. The paper also claims that their algorithm does not require an explicit simulator or oracle to run trajectories for a given policy for some number of rounds (as is usually the case with many existing RL methods). They further provide, regret guarantees for their algorithm and compare it with the existing approaches to show their usefulness and superiority.

**Questions:**

1) Each episode starts with a state drawn from a fixed distribution -- Seems to require a reset at the start of every episode? This seems to contradict the claim that their scheme is simulator-free. Am I missing something?
2) Since I am not very well versed with this topic, can the authors provide some insight on Assumption 2?

**Limitations:**

The authors have clearly stated the limitations of their work and the areas of future directions.

**Strengths And Weaknesses:**

+ Paper is technically solid.
+ The theoretical results appear to be sound and useful
+ The problem attacked is well-motivated

The main weakness of the paper according to me is readability.
- It might help to organize the outline of the proof in a more structured way by giving (sub-)headings for each the various parts, for easy parsing.

---

> ### Author Response · Authors · 2022-08-02
> **Response to Reviewer K6Mq**
>
> We appreciate your time and thoughtful evaluation of our paper. We recap your comment and present our detailed response as follows. We would also be happy to provide further clarifications if suitable.
>
> >*It might help to organize the outline of the proof in a more structured way by giving (sub-)headings for each the various parts, for easy parsing*
>
> - We have added the sub-headings in the revised manuscript (Please see Section 4.2 page 7, the changes are marked in blue).
>
> >*Each episode starts with a state drawn from a fixed distribution -- Seems to require a reset at the start of every episode? This seems to contradict the claim that their scheme is simulator-free. Am I missing something?*
>
>  - We consider an episodic setting which means that the state is reset after every H step. The state from which the new episode starts can be the same or the state can be drawn (by the environment) from a distribution.
>
> - Now, we clarify what simulator-free means in the literature. An algorithm is said to be simulator-based if it can access an oracle that allows the agent to query arbitrary state-action pairs and return the reward and the next state, hence greatly alleviating the intrinsic difficulty of exploration in RL. For example,  a naive exploration strategy that queries all state-action pairs uniformly at random already leads to the most efficient algorithm for finding an optimal policy [R1]. On the other hand, in a simulator-free algorithm, a trajectory is observed following the actions drawn from a policy.  We only observe the states, rewards, and utilities encountered in the trajectory.
>
> >*Since I am not very well versed with this topic, can the authors provide some insight on Assumption 2?*
>
> - Assumption 2 means that there exists a strictly feasible policy. Finding a strictly feasible policy is often easy. For example, consider the problem of minimizing the electric vehicle charging cost while maintaining a certain level of state of charge over a day (constraint). One naive policy can be to charge the vehicle to its fullest irrespective of the electricity price. This would certainly be a strictly feasible policy and one can measure $\gamma$. Hence, finding such a strictly feasible policy and computing $\gamma$ is often easy. We would also like to point out that our algorithm does not need to know the strictly feasible policy, rather, only needs the knowledge of $\gamma$ rather than a strictly feasible policy. Thus, if an oracle or expert provides the value  $\gamma$, i.e., the utility of a strictly safe value (one such value is enough), it would be enough.
>
> - Note that the assumption of the existence of such a strict feasible policy and knowledge of the value of $\gamma$ is common in practice [7], [1], [6],  and also in [R2] (just above section 3.2). The regret and violations both depend on $\gamma$ as well in those papers.  Theoretically, such a strict feasible policy is important to show strong duality and eventually bounding the dual variable. Strong duality has been proved in [R3] when the slater’s condition is satisfied. Note that the CMDP problem is non-convex in the policy form, thus, strong duality is not guaranteed in general. [1] uses the strong duality result to achieve the upper bound on the dual variable (Theorem 43, page 44). Note that if $\gamma$ is large, the upper bound becomes smaller (Lemma 1). Hence, finding a feasible policy that satisfies the constraint with a larger margin would reduce the regret bound. However, it would increase the violation (Theorem 1).
>
> [R1]. Azar, Mohammad Gheshlaghi, Rémi Munos, and Bert Kappen. "On the sample complexity of reinforcement learning with a generative model." arXiv preprint arXiv:1206.6461 (2012).
>
> [R2]. Chen, Liyu, Rahul Jain, and Haipeng Luo. "Learning Infinite-Horizon Average-Reward Markov Decision Processes with Constraints." arXiv preprint arXiv:2202.00150 (2022).
>
> [R3]. Paternain, Santiago, Luiz Chamon, Miguel Calvo-Fullana, and Alejandro Ribeiro. "Constrained reinforcement learning has zero duality gap." Advances in Neural Information Processing Systems 32 (2019).

---

> > ### Comment · Reviewer_K6Mq · 2022-08-08
> > **answers to my questions**
> >
> > I thank the authors for answering all my questions.

---

### Meta-Review · Area_Chair_1Rjt · 2022-08-28

**Recommendation:** Accept
**Confidence:** Certain

**Metareview:**

This paper considers minimizing the regret while learning a near-optimal policy in an episodic constrained MDP with linear function approximation. It proposes and analyzes a UCB-based algorithm proving sublinear regret.

The reviewers found the paper well-motivated and technically sound, and unanimously recommend acceptance. Please incorporate the reviewers' feedback in the final version of the paper. In order to strengthen the final paper, it would be helpful to:
- Incorporate toy experiments and empirically validate some of the paper's claims
- Include a discussion about the tightness of the upper bound

**Award:**

No

---

### Decision · Program_Chairs · 2022-09-14

Accept